# Empirical evidence on the efficiency of backward contact tracing in COVID-19

Joren Raymenants [1,2,4] ✉, Caspar Geenen [1,4], Jonathan Thibaut [1], Klaas Nelissen[1], Sarah Gorissen[1] & Emmanuel Andre[1,3]

Standard contact tracing practice for COVID-19 is to identify persons exposed to an infected person during the contagious period, assumed to start two days before symptom onset or diagnosis. In the first large cohort study on backward contact tracing for COVID-19, we extended the contact tracing window by 5 days, aiming to identify the source of the infection and persons infected by the same source. The risk of infection amongst these additional contacts was similar to contacts exposed during the standard tracing window and significantly higher than symptomatic individuals in a control group, leading to 42% more cases identified as direct contacts of an index case. Compared to standard practice, backward traced contacts required fewer tests and shorter quarantine. However, they were identified later in their infectious cycle if infected. Our results support implementing backward contact tracing when rigorous suppression of viral transmission is warranted.

Case-based interventions such as case isolation or contact tracing with quarantine have been crucial in controlling the ongoing COVID-19 pandemic, while reducing the need for indiscriminate contact reductions with high economic cost[1,2].

Contact tracing aims to identify and interrupt transmission chains by isolating infected patients and quarantining those at risk from infection. More infections are prevented, and epidemic control is improved, if the identification of patients and contacts at risk is rapid and comprehensive[3–6]. It has been a staple public health intervention in a variety of infectious diseases, notably sexually transmitted diseases and tuberculosis[7,8].

Worldwide investments in contact tracing programmes and research on the topic have not prevented repeated resurgence of community transmission of COVID-19, underscoring the urgent need for improved knowledge on the effective implementation of this key public health measure[6,9].

Forward contact tracing of an index case (the person diagnosed with COVID-19 undergoing contact tracing) intends to interrupt onward transmission from child cases (persons infected by the index case) by quarantining and/or testing contacts the index case has encountered during their infectious period[10–12]. In the light of substantial asymptomatic and pre-symptomatic transmission, the infectious period is generally assumed to start 2 days prior to onset of symptoms or diagnosis, whichever came first[13–18]. In addition to child cases, any practical forward tracing strategy probably identifies the parent case (the infector of the index case) and sibling cases (infected by the same parent case) some of the time, for example if the index case had repeated contact with their parent or sibling case during their own infectious period, or if the time from the index case's infection to their symptom onset or diagnosis was less than 2 days[12]. Forward contact tracing is the focus in most jurisdictions and has shown its ability to decrease COVID-19 transmission (Fig. 1)[13,14,19].

Backward contact tracing, or bidirectional contact tracing, which combines both approaches, specifically aims to identify the parent case and sibling cases by going back further in time[5,10–12]. In any practical implementation, additional child cases may also be identified through backward contact tracing, for example if the index case's infectiousness started more than 2 days before symptom onset[12].

Backward contact tracing is particularly promising in COVID-19 because a small proportion of index cases, the so-called superspreaders, generate the majority of secondary infections[11,20–27]. This phenomenon favours allocating resources to the identification of

[1]KU Leuven, Laboratory of Clinical Microbiology, Herestraat 49box 6711, 3000 Leuven, Belgium. [2]Algemene Interne Geneeskunde, UZ Leuven, Herestraat 49, 3000 Leuven, Belgium. [3]Laboratoriumgeneeskunde, UZ Leuven, Herestraat 49, 3000 Leuven, Belgium. [4]These authors contributed equally: Joren Raymenants, Caspar Geenen. ✉e-mail: joren.raymenants@kuleuven.be

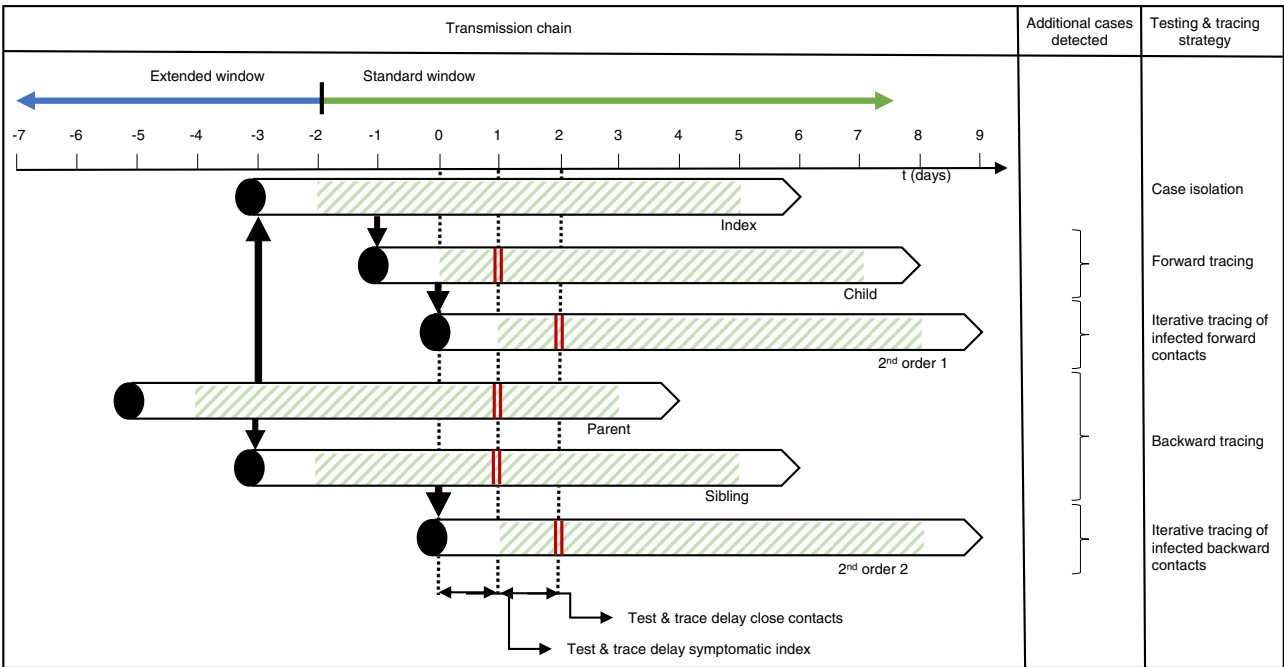

**Fig. 1 | Schematic representation of the different testing and tracing strategies and which parts of the chain of transmission they can uncover.** On the left, a transmission chain is shown where COVID-19 spreads from a parent case to an index case and their sibling case at a shared source event. The index, sibling and child cases all spread their infection further. Black arrows show transmission events, while green diagonals show the infectious period of each case. The index case develops symptoms on day 0 and gets tested as soon as possible. Double full vertical lines highlight when each case is detected as a contact, considering a combined testing and tracing delay of 1 day and testing of identified contacts as soon as possible. The standard and extended contact tracing windows are shown above the timeline. The testing and contact tracing strategies and which additional cases they identify are shown on the right. As especially the parent case demonstrates, a possible drawback of backward contact tracing is that some infected contacts are detected at a later stage of their infection, decreasing the effectiveness of testing and quarantine measures. It must be noted that the directionality of transmission and thus the position of an infected individual in the transmission tree is usually difficult to ascertain in practice.

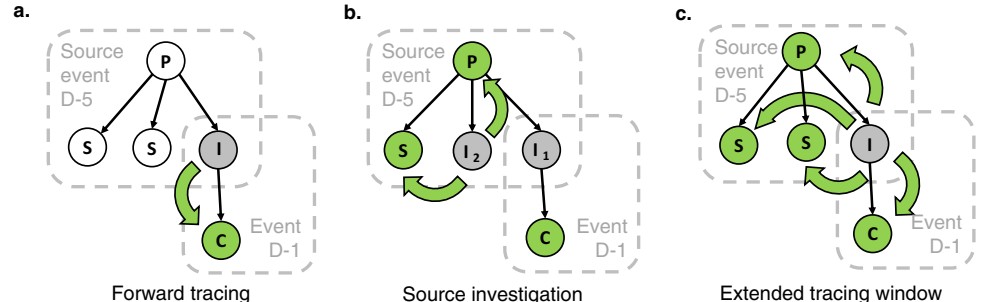

**Fig. 2 | Schematic representation of contact tracing strategies.** Thin black and thick green arrows indicate the directions of transmission and contact tracing respectively. I index case, C child case, P parent case, S sibling case. White circle: undetected case. Grey circle: case detected through symptomatic screening. Green circle: case detected through contact tracing (**a**) When an index case is diagnosed, the child case at event D-1 is identified through standard forward tracing. A source investigation would fail at this stage, because there is no indication of further infections at the source event. **b** Source investigation does succeed when a second index case $I_2$ is diagnosed independently of the initial index case $I_1$. As the source event becomes clear due to identification of multiple infections, all attendants are traced. **c** An extended tracing window quickly identifies parent, sibling and child cases as direct contacts of the first index case.

source cases and events, as a high rate of infection can be expected amongst individuals exposed to the same source. Endo et al. estimate bidirectional contact tracing to result in two to three times the number of subsequent cases averted compared to forward contact tracing alone in a simple branching model for COVID-19[10]. Kojaku et al. show backward contact tracing to be highly effective in terms of the number of prevented cases per quarantine when running an Susceptible-Exposed-Infectious-Removed model on synthetic and empirical contact networks, even if contact tracing comprehensiveness is low[11].

One potential difficulty of backward contact tracing lies in the inherent delays involved in testing, tracing and quarantine—where

infected contacts who are sibling or parent cases risk being detected after or near to the end of their infectious period[3,18]. This could reduce efficiency and increase the relative cost of testing and quarantine (Fig. 1). Due to these delays, immediate testing of identified contacts in support of iterative tracing may be especially relevant in backward contact tracing.

The real-world implementation of backward contact tracing can be broadly subdivided into a source event approach and an extended contact tracing window approach (Fig. 2).

Several countries have rolled out an approach focusing on source events, which are events where the index case is suspected to have

contracted COVID-19. The identification of such an event leads to the screening of attendants at risk, which usually includes more individuals than the direct contacts of the index case under investigation[28–32]. This is because the risk at these events is not related to the index case, but to an unknown parent case. High positivity rates (PR) have been reported for attendants of some source events[33]. In practice, this approach is usually reliant on the identification of multiple confirmed or probable infected cases at the same event, for example by pooling of contact tracing data from different index cases or asking the index case about other cases in their environment. As a result, the approach can fail to identify the source event at the time of identification of the initial index case.

Another approach is to extend the contact tracing window back in time and to systematically refer all close contacts for quarantine and/or testing (Figs. 1 and 2). This assumes that, if the tracing window is extended backward by at least the incubation period of the index case, the parent case can be identified, as well as sibling cases present at a shared source event. To this end, the contact tracing window should be extended far enough to include most of the variability in incubation periods[34].

Several modelling studies underscore the benefits of extending the contact tracing window for COVID-19. Bradshaw et al. show in a stochastic branching process model that extending the contact tracing window from 2 to 6 days before onset or diagnosis improves the reduction in the effective reproduction number by 85 to 275% when using manual contact tracing only (performed by humans rather than through digital means)[12]. Their findings are robust to contextual factors such as case ascertainment rate, test sensitivity, basic reproduction number and the percentages of asymptomatic, pre-symptomatic and environmental transmission. Fyles et al. also show in a branching process model that an extended contact tracing window results in a linear decrease in the growth rate up until around 8–10 days prior to symptom onset or diagnosis, although additional gains are highly sensitive to recall decay[5].

Whilst there is evidence from modelling studies pointing at the potential benefits of backward contact tracing, no study has evaluated the efficiency in practice. The PR of screened contacts has been proposed as an indicator for efficient allocation of testing and quarantine[35,36]. In this cohort study, we thus determined the PR of additional close contacts (for the purpose of this article this includes co-attendants of high-risk events of up to 20 persons) identified in an extended contact tracing window, starting 7 days before onset of symptoms or diagnosis, whichever was earlier. This window was chosen to include the source event most of the time[32–34]. We tested the hypothesis that the PR amongst additional contacts in the extended tracing window would be at least as high as amongst a control group of patients attending the test centre for symptoms suggestive of COVID-19. In a first subgroup analysis, we explored how far back the contact tracing window should extend, by calculating the PR of identified contacts grouped by day of last exposure. Our second hypothesis was that the risk would not be limited to possible source events identified at the time of the tracing interview. Therefore, the second subgroup analysis compared our strategy to a source investigation approach, by subgrouping contacts last exposed in the extended contact tracing window according to presence at suspected source events.

## Results

### Study cases and contacts

Our test and trace programme ran from September 2020 until May 2022. Due to gradual improvements in organisation and data collection, there was a marked increase in the ratio of contacts with outcome data after the initial months of the programme (Supplementary Fig. 3). The study period for the main analyses was chosen from 1st February 2021 to 31st May 2021, which was after the initial set-up phase of the

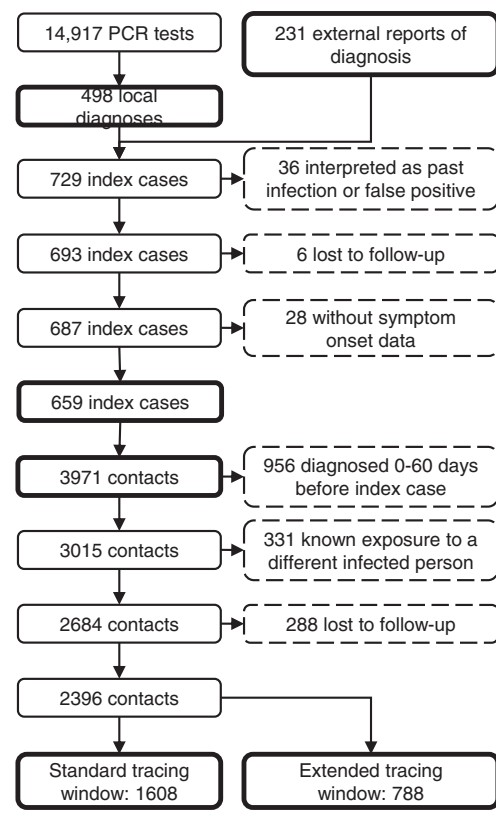

**Fig. 3 | Inclusion flowchart main study period.** The number of included and excluded cases and contacts is shown for the main alpha-dominant cohort.

programme and included both an upward and a downward trend in country-wide infection rates.

In total, 14,917 students underwent RT-qPCR testing at our centre in this period (3.8 tests per 1000 persons daily), resulting in 498 students with a new diagnosis of COVID-19. A further 231 positive RT-qPCR test results of students in the study population were reported to us from external sources, resulting in a total of 729 cases. Thirty-six (4.9%) of these were interpreted as a past infection or false positive by the treating physician, leaving 693 actual cases (14-day incidence of 245 per 100,000). Six cases (0.9%) were considered lost to follow-up, because they could never be contacted by the contact tracing team, and 28 (4.1%) were excluded because data on presence of symptoms was missing. Therefore, 659 index cases remained in the analysis (Fig. 3).

In total, 72.5% of index cases self-reported being symptomatic at the time of testing, which was similar to the national average[37]. Index cases had a mean age of 21.4 years (SD: 3.60 years, missing data 15.0%) and were 51.1% male (missing data 12.1%).

Contact tracing of the index cases resulted in 3971 case-contact pairs (mean 6.0 contacts per case, 2.2 times the national average[37]), of which 956 (24.1%) were excluded because the contact person already had a positive test result 0 to 60 days before the positive test of the index case. Another 331 (11.0%) contacts were excluded because they already had a known exposure to a different infected individual within 7 days before the tracing interview. Finally, 288 contacts (10.7%) were lost to follow-up. The distribution of the number of contacts per index case in shown in Supplementary Fig. 4.

The resulting 2396 contacts were divided into two groups. The standard tracing window group, which would have been identified through standard practice, consisted of 1608 individuals in close contact with the index case in the period from 2 days before onset or test until the contact tracing interview. The backward traced group consisted of 788 additional contacts in the extended tracing window,

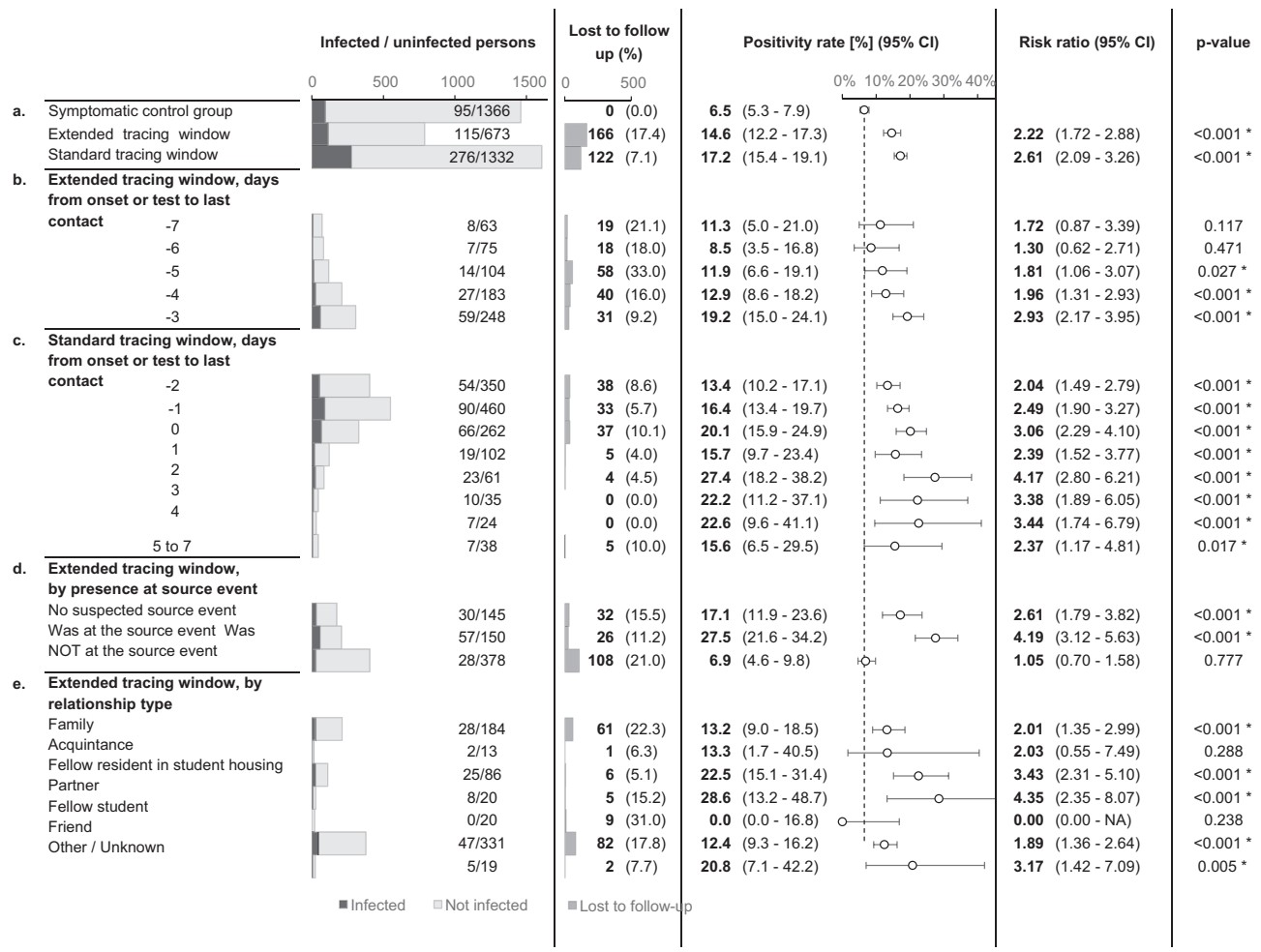

**Fig. 4 | Outcomes, positivity rates and risk ratios for contacts of index cases with corresponding *p* values.** The dotted line indicates the positivity rate in the control group. The error bars indicate 95% two-sided confidence intervals (Clopper–Pearson). * indicates a statistically significant difference in comparison to the control group (*p* < 0.05) as assessed using a two-sided Chi-squared test, not adjusted for multiple comparisons. Section **a** tests the main hypothesis by comparing the extended tracing window to the symptomatic control group. Subgroups by the numbers of days from onset or test of the index case to the last interaction with the index case are shown in section **b**, **c** for the extended and standard tracing windows respectively. Section **d** shows subgroups according to presence at suspected source events, and subgroups by relationship type are shown in **e**.

i.e. their last close interaction with the index case was 3 to 7 days before onset or test. For the main analysis, we did not make assumptions on the directionality of transmission. Therefore, both the forward and backward traced group likely included parent, sibling and child cases.

We did not collect demographic data on contacts of index cases.

The control group consisted of all 1461 students who attended our test centre for the first time with self-reported symptoms suggestive of COVID-19 as the main reason for their test.

There was a slightly higher percentage of women in the control group (56.5%, missing data 3.0%) compared to the index cases, while the mean age was similar (22.0 years; SD 3.84 years, missing data 3.0%). The temporal distribution of individuals in the backward traced contact and symptomatic control groups is shown in Supplementary Fig. 1a.

### High risk of infection in the extended tracing window
By extending the contact tracing window, 49% more contacts at risk and 42% more cases were identified as direct contacts of an index case, compared to standard contact tracing practice alone.

The risk of infection in the standard and extended tracing window groups was similar, namely 17.2% in the former (CI 15.4–19.1%) and 14.6% in the latter (CI 12.2–17.3%). The risk in the extended

tracing window group was significantly higher (risk ratio 2.22, CI 1.72–2.88, *p* < 0.0001) than the risk of 6.5% (CI 5.3–7.9%) in the control group, demonstrating the relative efficiency of extending the contact tracing window to 7 days prior to symptom onset or test (Fig. 4).

Contacts in the standard and extended tracing window groups were subgrouped by their last day of contact with the index case, relative to symptom onset or test. The results show that the number of additional identified close contacts per day decreased markedly as the tracing window was extended backward. The risk of infection varied from 8.5 to 19.2%, and the confidence interval lower bound did not drop below 3.5% for any of these subgroups in the extended tracing window. For day 3, 4 and 5 before onset or test, the risk was significantly higher than the control group (*p* < 0.05).

### The risk is not limited to suspected source events
An important consideration when deciding between a source investigation approach and an extended tracing window is the risk of infection for contacts not present at suspected source events. A suspected source event was identified for 80.6% of index cases. If the contact tracing interview failed to suggest a source event, the risk of infection for extended tracing window contacts was 17.1% (CI 11.9–23.6%). If a source event was identified, the risk was around four times higher for

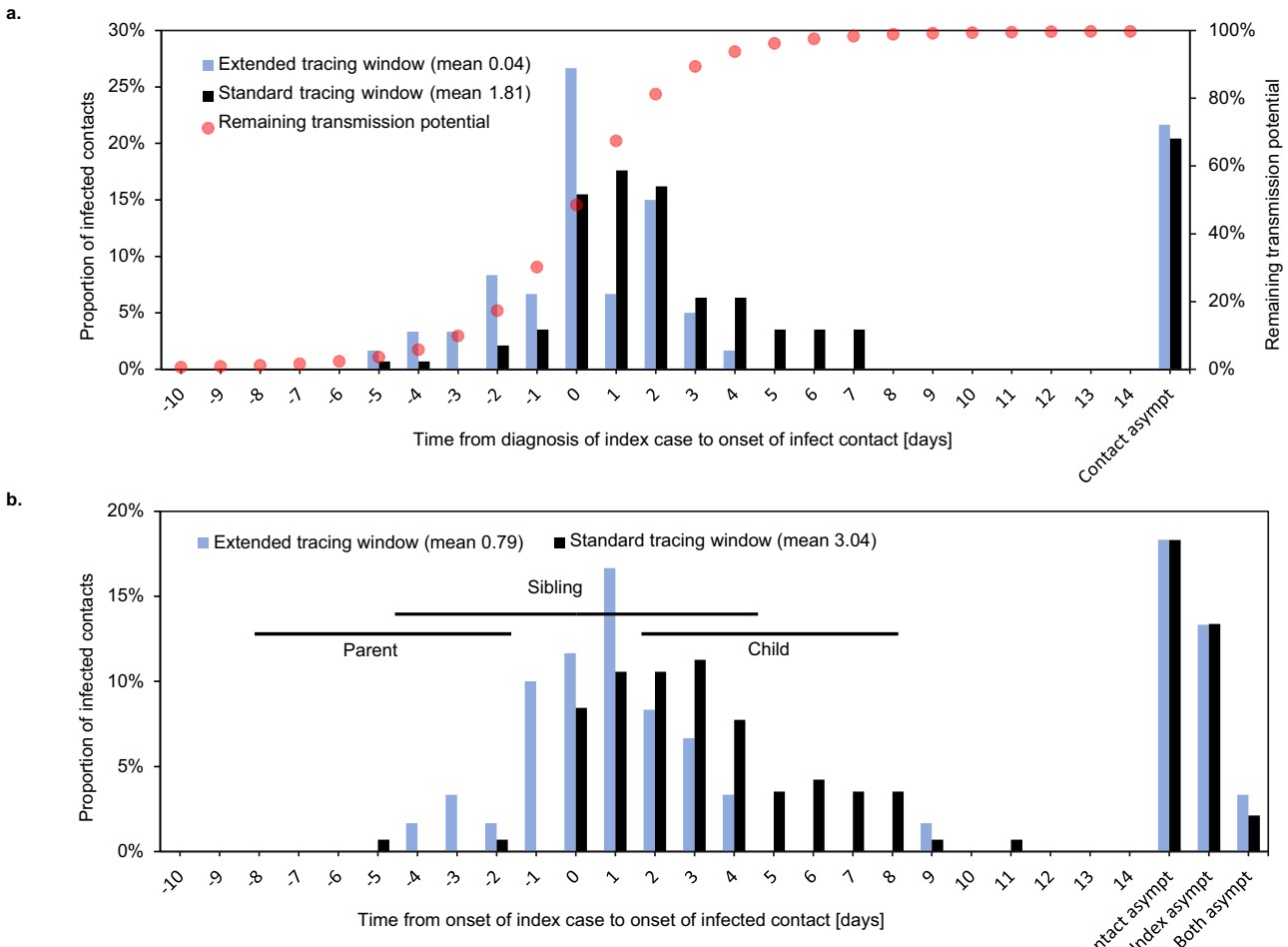

**Fig. 5 | Infection stages of COVID-19-positive contacts.** Symptom onset in infected contacts relative to sampling (**a**) or symptom onset (**b**) of the index case. Asympt asymptomatic. Case–contact pairs that were excluded from the mean calculation because either or both were asymptomatic, are shown on the right. Panel **a** shows the delay between detection of an index case and symptom onset of their infected contact. Red dots show the estimated remaining fraction of transmission potential of the infected contact at the time of sampling of the index case. Forward traced symptomatic contacts were detected on average 1.8 days earlier in their infectious cycle than their backward traced counterparts, assuming equal delays between index case diagnosis and tracing of the contact. This resulted in a 28% lower mean remaining transmission potential for backward traced contacts at the time of index case testing. Panel **b** shows the delay between symptom onset of an index case and their infected contact. Horizontal lines indicate the 25th–75th percentile ranges of expected timings for parent, sibling and child cases, based on a published normal distribution of the serial interval[38]. The observed timings are compatible with a high proportion of sibling cases and few parent or child cases in the backward traced group.

contacts who attended the event (absolute risk 27.5%, CI 21.6–34.2%) compared to those who did not. The latter group still had a risk of 6.9% (CI 4.6–9.8%), which was similar to the symptomatic control group but not significantly higher (Fig. 4a).

### Risk by relationship type

In an explorative subgroup analysis, extended tracing window contacts were grouped according to relationship type with the index case. The majority of identified contacts were either family (28.6%), fellow residents in student housing (12.3%), or friends (48.2%). Each of these three groups had a significantly increased infection risk as compared to the symptomatic control group. The other subgroups lacked sufficient numbers for statistical power (Fig. 4e).

### Backward contact tracing identifies cases later in their infection

Effective contact tracing requires the detection of infected contacts as soon as possible, before they reach the end of their contagious period. The sibling and especially parent cases targeted by backward contact tracing can be expected to be in a later stage of infection compared to forward traced contacts, potentially leading to lower efficiency of tracing, testing and quarantine measures.

Indeed, when comparing the date of detection of an index case with the onset date of their infected contact, the infected contacts in the extended tracing window were on average 1.8 days later in their infectious cycle compared to those in the standard tracing window (Fig. 5a). The difference could be interpreted as a reduction in contact tracing efficiency equal to an additional testing or tracing delay of the same period.

The difference of 1.8 days in contact symptom onset relative to index case detection is much smaller than we would expect if all backward and forward traced contacts were parent and child cases, respectively (double the mean serial interval of around 5 days)[38]. One possible explanation is than sibling cases make up a considerable share of contacts in both groups. Although we cannot ascertain the relative positions of infected contacts in the transmission tree, the observed timings would be consistent with a higher fraction of sibling cases in the backward traced group and a minority of parent cases in both groups (Fig. 5b).

To quantify the fraction of transmissions averted through quarantine of symptomatic infected contacts, we used a distribution of timing of transmission relative to symptom onset (Fig. 5a)[39]. At the time of testing of the index case, the mean fraction of remaining

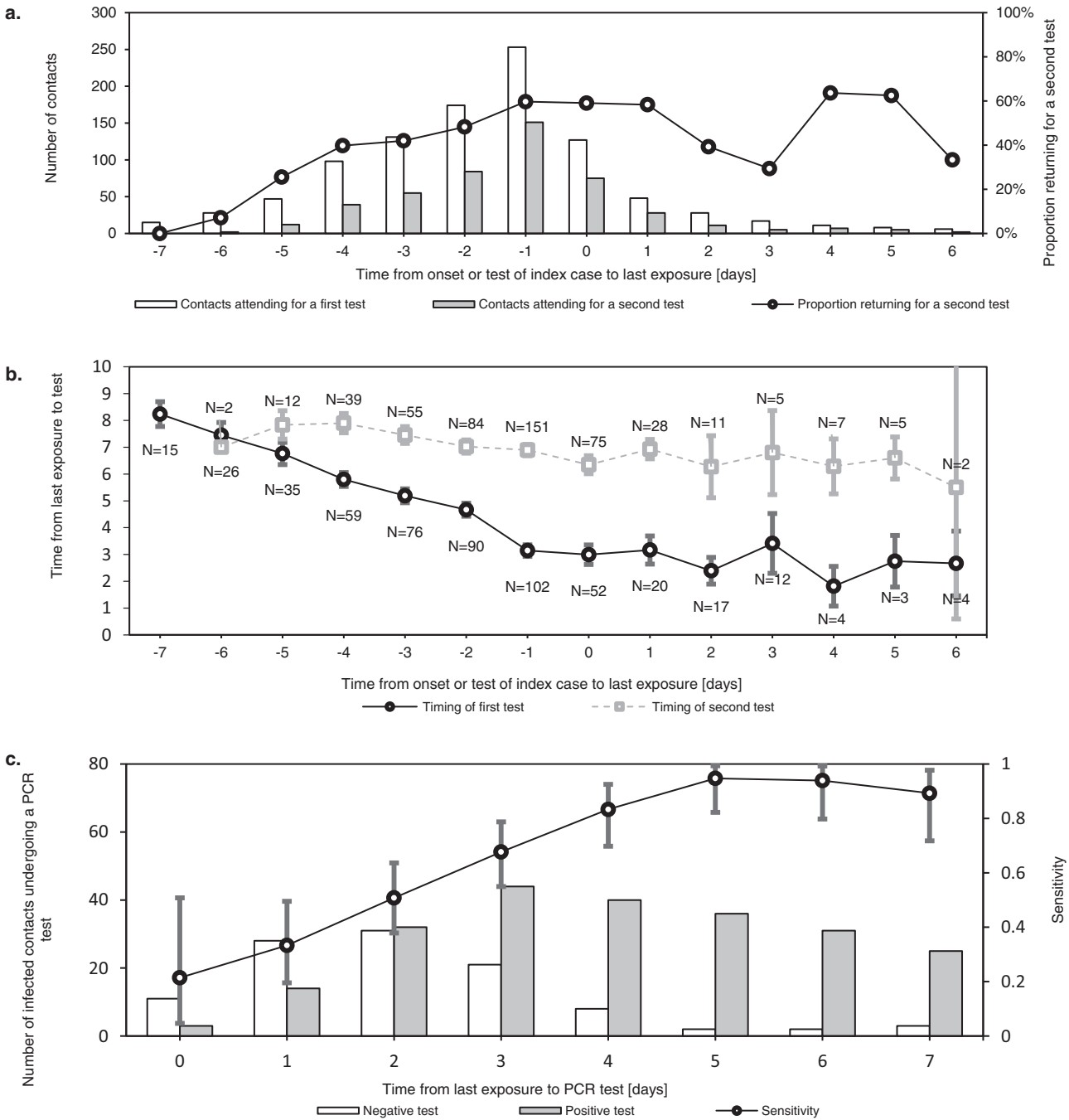

**Fig. 6 | Timing of RT-qPCR testing in contacts as performed in the study period and the diagnostic accuracy of such tests by day since last exposure.** Error bars indicate 95% confidence intervals. **a** shows the number of contacts who underwent a first and second tests at our test centre after their exposure. This demonstrates how testing immediately after exposure ("test to trace") was most often complemented with testing after a latent period ("test to release"). While the former mainly supports iterative tracing and in some cases a shortened isolation period, the latter allows shortening of quarantine for non-infected contacts. As the delay between last exposure and symptom onset or testing of the index case increased, the percentage of contacts requiring two tests decreased. **b** shows the mean timing of first and seconds tests at our centre for contacts, relative to their last exposure. The difference in timing of the first and second tests is reduced as the contact tracing window is extended further back in time. **c** shows the test results of infected contacts by day after last exposure, demonstrating how the sensitivity of RT-qPCR testing increased rapidly in the first days after exposure.

transmission potential was 28% lower for infected backward as opposed to forward traced contacts.

**Less tests and shorter quarantine in the backward traced group**
The value of contact testing depends not only on test specific diagnostic performance, but also on timing. Immediate testing after contact identification can accelerate iterative tracing ("test to trace"). It can also reduce the total duration spent in quarantine and isolation, in settings where release from isolation is dependent on the timing of diagnosis. Tests after a latent period are more sensitive and can thus be used to allow shortening of quarantine for non-infected contacts ("test to release") (Fig. 6c)[40].

During the study period, contacts were requested to undergo RT-qPCR tests both as soon as possible after identification and again 7 days after last exposure, which is reflected in the timing of contact testing in our dataset (Fig. 6b).

As backward traced contacts were detected a minimum of 3 days after their last exposure by definition and an average of 4.0 days longer after last exposure than forward traced contacts in our dataset, a single test at identification was more likely to serve both a "test to trace" and "test to release" strategy concurrently. We estimate a reduction of 17% in the number of tests required per traced contact, based on a delay from index case testing to contact testing of 1 day (Supplementary Fig. 5).

Another consequence of this inherent difference in last exposure date is that, in our dataset, the mean duration of quarantine was 3.0 days (57%) shorter for contacts in the backward traced group compared to the forward traced group. This result assumes a duration of quarantine from index case diagnosis until 7 days after exposure, with a minimum of 1 day to allow for contact testing (Supplementary Fig. 5).

### Impact of changing viral variants

Consecutive SARS-CoV-2 variants of concern (VOC) may have challenged the effectiveness of contact tracing in several ways. First, increased intrinsic transmissibility may have rapidly overwhelmed the public health system[41,42]. Second, shortened incubation periods and serial intervals possibly outpaced the delays inherent in testing and tracing[34,43,44]. To assess the influence of these altered transmission dynamics, the main analysis was repeated for periods when the Delta and Omicrons VOCs were dominant nationally (Fig. 7). These periods differed from the main study period not only in terms of the dominant circulating VOC, but also in the immune status of the target population, the general contact restrictions in place, the COVID-19 incidence rate and the government requirements concerning testing and quarantine (Supplementary Figs. 3, 7, 9 and 10)[37,45,46].

Unfortunately, follow-up rates dropped markedly after the main study period, especially for contacts in the extended tracing window. During the periods characterised by Delta dominance, backward traced contacts had similar PR to both forward traced contacts and symptomatic controls, further supporting our main hypothesis (Fig. 7b). During the periods characterised by Omicron dominance and an almost fully vaccinated population, backward traced contacts retained a very high PR (mean 13.3%, CI 8.5–19.5%)[47,48]. It was however significantly lower than the much increased PR in symptomatic controls and forward traced contacts (Fig. 7c).

### Iterative contact tracing in a branching process model

As mentioned, iterative contact tracing of infected contacts is thought to play a larger role in backward contact tracing. However, many of the reported contacts in our dataset were outside the study population, which means their contacts were not iteratively traced using the same backward tracing strategy if infected. To estimate how efficient backward contact tracing would be if all infected contacts were iteratively traced, we used a simple branching process model. The design of the model, described in Supplementary Methods, requires no assumptions on the direction of transmission or the probability of an infected contact being traced. This model allowed us to estimate, for our setting, the total expected number of traced contacts from a primary index case, over multiple iterations of contact tracing, based on the observed numbers of infected contacts in the main study period (Supplementary Fig. 8 and Supplementary Table 1). We then quantified several measures of costs and benefits of an extended contact tracing window relative to standard contact tracing practice alone.

The results are summarised in Fig. 8 and model details are shown in Supplementary Fig. 8. When taking into account iterative tracing of backward and forward traced contacts, an extended contact tracing window identified 55% more cases than forward tracing only (Fig. 8a). It also detected 61% more asymptomatic cases and averted 38% more infections, using the measure of remaining transmission potential described above. On the other hand, backward tracing required 78%

more contacts to be traced, 67% more tests and 40% more quarantine days (Fig. 8b). Additional benefits and costs both declined for each day the contact tracing window was extended backwards. Although fewer cases were identified per traced contact, the lower number of required tests and quarantine days lead to a cost-benefit balance which remained favourable relative to forward contact tracing, depending on which cost and benefit measures were considered (Fig. 8c).

## Discussion

This study lays out a strategy for backward contact tracing which markedly improves the effectiveness of contact tracing in the setting of COVID-19. It identified an additional 42% (or 55% in a mathematical model of iterative tracing) of cases not detected through the contact tracing protocol used in most jurisdictions, gains which are likely to have a major impact on epidemic control[12]. The main trade-off was that infected backward traced contacts were identified on average 1.8 days later in their infectious cycle than forward traced contacts. However, the burden of testing and quarantine was lower in backward traced contacts due to inherent differences in the timing of their last exposure to the index case. Our results contradict perceptions on cost efficiency, which continue to hamper the broader introduction of backward contact tracing as a standard mitigation strategy.

Our approach was to extend the contact tracing window back in time from 2 to 7 days before symptom onset or test, and to systematically refer all identified close contacts in this period for testing, as well as co-attendees of small high-risk events. This simple change in standard protocol, which could be implemented both in manual and digital contact tracing, apparently allowed sibling cases to be identified quickly as direct contacts of the index case.

Our data show that 49% more direct contacts were reported when extending the contact tracing window by 5 days. As the contact tracing window was extended backward, fewer additional contacts were identified per day. This could be explained by recall decay, but also by recurring contacts with the same individuals. Household contacts, for example, were often excluded from the backward tracing group because they were also exposed in the standard contact tracing window.

Crucially, contacts last encountered during the extended tracing window had a higher risk of testing positive compared to symptomatic patients in the same population. These results were independent of whether they were friends, family or fellow residents of the index case.

PR amongst symptomatic individuals are dependent on many factors, such as the level of community transmission of SARS-CoV-2 and other respiratory viruses. Still, this group was chosen as a control group, because it represents a high bar and testing of symptomatic patients is standard in most protocols globally[45–48].

Unfortunately, we were unable to replicate the high follow-up rates of the main study period in subsequent periods with different dominant VOC. We attribute this mainly to gradual loosening of government-mandated testing protocols and higher viral circulation, forcing the contact tracing team to prioritise contact notification over follow-up[45,46] (Supplementary Fig. 3). The control group probably also suffered a further reduction in reliability after the main study period, due to the rollout of alternative testing methods such as pharmacy-based and self-administered rapid antigen tests and the progressive scaling back of RT-qPCR testing in general[37,45,46]. Based on follow-up rates, we chose four subsequent periods of interest, characterised by Delta and Omicron VOC dominance, for analysis (Supplementary Figs. 3, 6 and 7). These periods also differed from the main study period with regards to several other factors, such as general contact restrictions, population immunity and government test and quarantine strategy (Supplementary Figs. 9 and 10)[37,45,46]. In the Delta periods, the PR of backward traced contacts was similar to the symptomatic control and forward traced groups. In the Omicron periods, it was significantly lower than that of symptomatic and forward traced

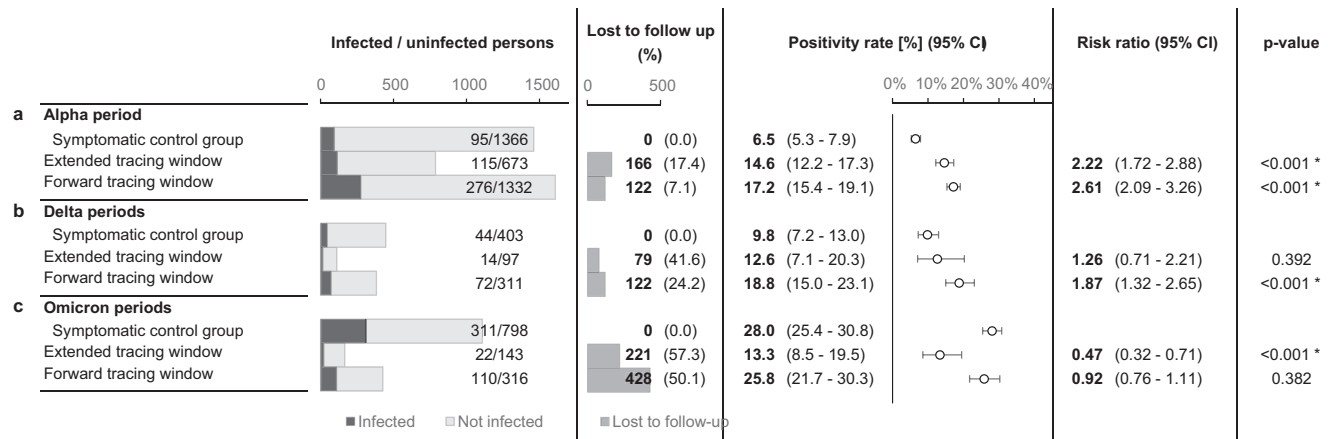

| | Infected / uninfected persons | Lost to follow up (%) | | Positivity rate [%] (95% CI) | | Risk ratio (95% CI) | p-value |
|---|---|---|---|---|---|---|---|
| **a Alpha period** | | | | | | | |
| Symptomatic control group | 95/1366 | 0 | (0.0) | 6.5 (5.3 - 7.9) | | | |
| Extended tracing window | 115/673 | 166 | (17.4) | 14.6 (12.2 - 17.3) | | 2.22 (1.72 - 2.88) | <0.001 * |
| Forward tracing window | 276/1332 | 122 | (7.1) | 17.2 (15.4 - 19.1) | | 2.61 (2.09 - 3.26) | <0.001 * |
| **b Delta periods** | | | | | | | |
| Symptomatic control group | 44/403 | 0 | (0.0) | 9.8 (7.2 - 13.0) | | | |
| Extended tracing window | 14/97 | 79 | (41.6) | 12.6 (7.1 - 20.3) | | 1.26 (0.71 - 2.21) | 0.392 |
| Forward tracing window | 72/311 | 122 | (24.2) | 18.8 (15.0 - 23.1) | | 1.87 (1.32 - 2.65) | <0.001 * |
| **c Omicron periods** | | | | | | | |
| Symptomatic control group | 311/798 | 0 | (0.0) | 28.0 (25.4 - 30.8) | | | |
| Extended tracing window | 22/143 | 221 | (57.3) | 13.3 (8.5 - 19.5) | | 0.47 (0.32 - 0.71) | <0.001 * |
| Forward tracing window | 110/316 | 428 | (50.1) | 25.8 (21.7 - 30.3) | | 0.92 (0.76 - 1.11) | 0.382 |

■ Infected　■ Not infected　■ Lost to follow-up

**Fig. 7 | Outcomes, positivity rates and risk ratios for contacts of index cases with corresponding *p* values.** The contact took place in selected periods, differing with regards to the dominant variants of concern, immunity, level of viral circulation, social contact restrictions and government testing/quarantine strategy. The error bars indicate 95% two-sided confidence intervals (Clopper–Pearson). * indicates a statistically significant difference in comparison to the control group (*p* < 0.05) as assessed using a two-sided Chi-squared test, not adjusted for multiple comparisons. **a** repeats the main study outcomes from Fig. 4a, while the results from subsequent periods are shown in **b**, **c**.

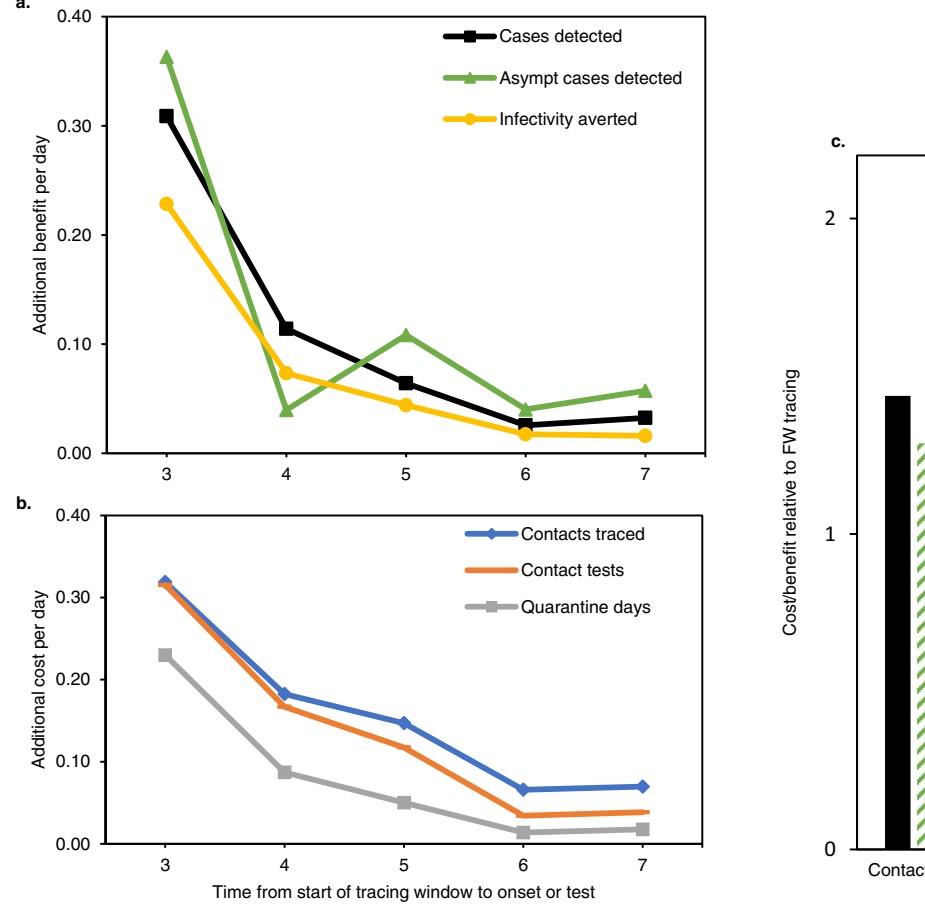

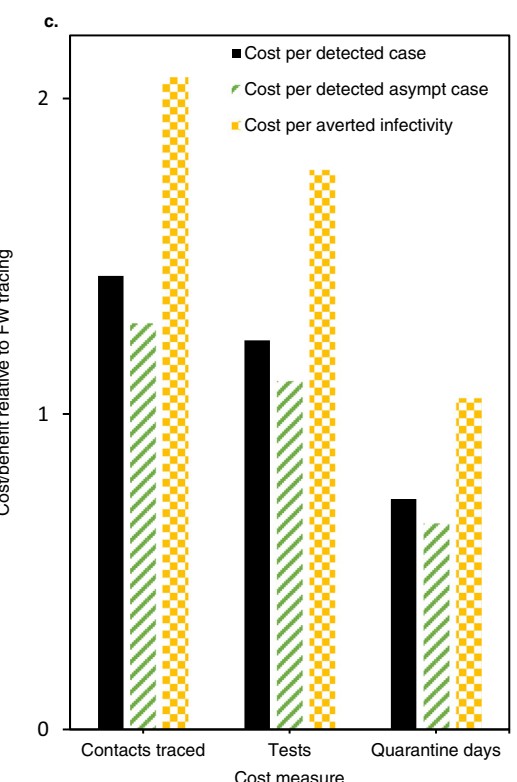

**Fig. 8 | Cost-benefit analysis of backward versus forward contact tracing in our setting, using a simple branching process model of iterative contact tracing.** Asympt asymptomatic. **a**, **b** show the marginal benefits and costs respectively, per day that the tracing window is extended backward. Both are given as a fraction of the benefits and costs of a standard forward tracing window. **c** show the total cost/benefit ratio of a contact tracing window extended to 7 days before onset or test, relative to a standard forward tracing window. Combinations of three cost and three benefit measures are shown. "Averted infectivity" denotes the number of detected cases, multiplied with their remaining fraction of transmission potential according to Fig. 5a. This measure of benefit accounts for the observation that backward traced cases were detected later in their infectious cycle. In this figure, "averted infectivity" can be considered equivalent to the number of averted infections, with the important caveat that it only includes child cases of a detected case, not any subsequent averted branches of the transmission tree.

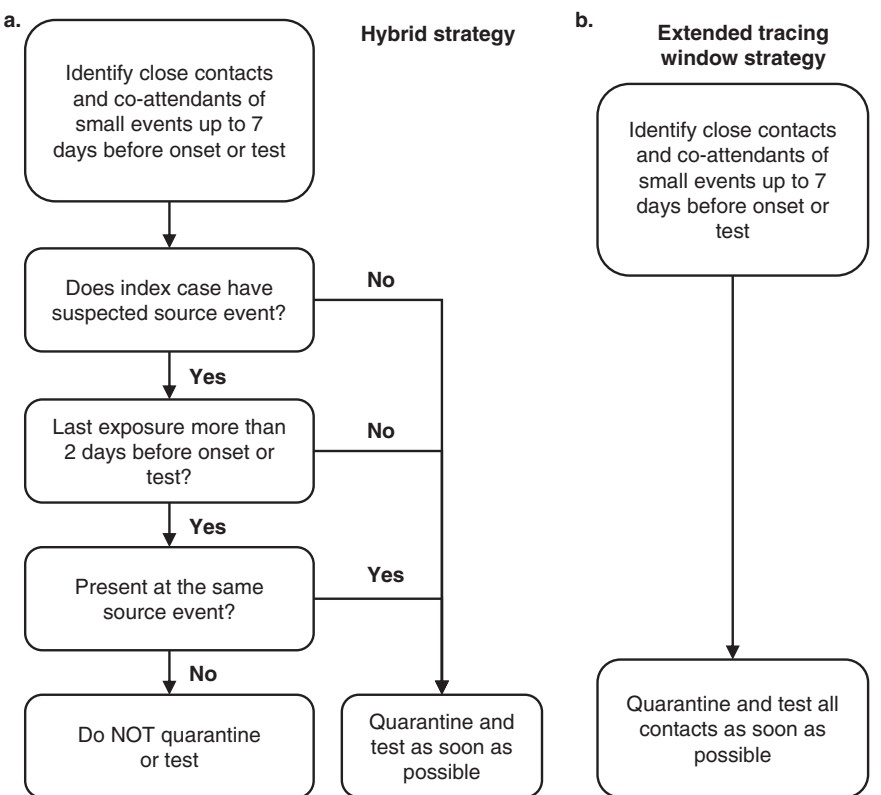

**Fig. 9 | Schematic representation of two possible strategies for backward contact tracing, based on our results. a** shows a hybrid strategy, which avoids testing contacts in the extended tracing window who were not present at the suspected source event. **b** shows an extended tracing window strategy with systematic testing of all contacts.

reference groups (Fig. 7). However, the lower bound of the PR of backward traced contacts remained above the threshold PR of 5 and 4% that the World Health Organisation (WHO) and European Centre for Disease Control (ECDC) recommended as a target indicator for comprehensive testing, when considering all tests performed in a population[47,48]. It should be noted that we did not adjust the range of the extended tracing window to accommodate shorter incubation periods and serial intervals reported for the Delta and Omicron VOCs[34,43,44].

The high PR observed in contacts last seen before the contagious period can be explained by several mechanisms. First, the index case may have become contagious more than 2 days before symptom onset or test. However, the inferred number of child cases in the backward traced group was low. Second, the source case is likely to be amongst earlier contacts, but parent cases also likely formed a minority in this group. Third, due to a proven individual propensity to shed live virus and an above average number of social interactions, the source case could have initiated other infections among the index's contacts[11,49]. This explanation is supported by previous reports on the role of superspreading in COVID-19 transmission[20,22,23]. The finding that backward traced contacts seem to more likely consist of sibling rather than parent or child cases, also supports this mechanism (Fig. 5). Fourth, more distant relatives in the transmission tree could be detected due to wider circulation in an index case's social circle. Fifth, recall decay may cause index cases to forget contacts with whom they had shorter, fewer and less close interactions. There may also be a more intentional tendency to mention only those contacts who the index case considers at risk.

A question that arises is whether it is worthwhile to quarantine and test a contact in the extended tracing window, if a source event was identified which the contact at hand did not attend. Our results show that the risk of infection for such a contact (6.9%, CI 4.7–9.7%) was only

a quarter of that of source event attendees, but still similar to the symptomatic control group. It was also higher than the WHO and ECDC targets of 5 and 4% mentioned above[47,48].

These results speak in favour of simply referring all close contacts in the extended tracing window for testing and quarantine, even if they were not present at the suspected source event. Alternatively, jurisdictions favouring the implementation of a source investigation strategy would do well to switch to an extended contact tracing window approach when no clear source event is identified at the time of the contact tracing interview (hybrid strategy, Fig. 9).

Previous studies have emphasised that the benefits of backward contact tracing hinge on the ability to identify first the parent case and then sibling cases in a two-step process, which is likely to be highly susceptible to testing and contact tracing delays[5,10–12]. However, the distribution of differences in symptom onset dates between index cases and their backward traced contacts suggests that most backward traced infected contacts may have been sibling cases identified as direct contacts of the index case, without the need to first identify the parent case (Fig. 5b). This inference of relative positions in the transmission tree should be interpreted with caution. The observed differences in onset time between index cases and their contacts are dependent on a priori probabilities of contacts being sibling, parent or child cases. These probabilities are in turn dependent on the reproduction number, overdispersion of the offspring distribution and probabilities of sibling cases being direct contacts of an index case.

Backward traced contacts were detected 1.8 days later in their infectious cycle, compared to forward traced contacts. We estimated that this later detection of infected backward traced contacts was associated with a 28% reduction in the fraction of infectiousness remaining at the time of testing of their respective index case.

We would argue that this effect is compensated for by a lower testing and quarantine burden for backward traced contacts.

Compared to forward traced contacts, the last exposure of backward traced contacts to the index case was 4.0 days earlier. In our setting, this reduced the mean duration of their quarantine by 3.0 days and often eliminated the need for two tests (Fig. 6 and Supplementary Fig. 5).

In both backward and forward traced contacts, the rapidly increasing test sensitivity in the first days after exposure supports the implementation of an initial "test to trace" immediately after identification, which accelerates iterative tracing and can expedite release from isolation, where this is dependent on the time of testing (Fig. 6 and Supplementary Fig. 5). A "test to release" after a latent period can have sufficient sensitivity to end the quarantine of uninfected contacts. As backward traced contacts are, by definition, identified late after exposure to the index case, a "test to trace" and "test to release" can be combined into a single test more often than in their forward traced counterparts (Supplementary Fig. 5).

To take into account the effects of iterative tracing of backward and forward traced infected contacts, we built a simple branching process model. The model showed that, relative to forward iterative tracing only, backward iterative tracing identified 55% more cases, which is within the broad range of values estimated by Endo et al., at a cost of tracing 78% more contacts and performing 67% more tests[10]. Although in this model the remaining fraction of the infectious period was 30% lower for backward traced contacts, the cost-benefit balance remained favourable, when considering quarantine days as the main cost.

Overall, our results show that the immediate cost and burden of backward contact tracing can be proportional to the benefits of additional detected cases and averted transmissions. Our data do not allow inferences about the impact of backward contact tracing on the effective reproduction number, which can be considered proportional not to averted cases but to unaverted cases, an unknown in this study. However, several modelling studies have suggested that the improved epidemic control offered by backward contact tracing has the potential to dramatically lower costs to society, in the form of reduced testing, quarantine and illness[5,10–12].

The study has several limitations. First, the main analyses took place in the setting of moderate general contact restrictions, which altered social patterns significantly and likely increased the efficiency of identifying source individuals by decreasing the number of contacts in general and casual contacts in particular, which are harder to identify through manual contact tracing. Second, index cases were young adults in tertiary education, whose socio-economic status and contact patterns may differ significantly from other age and social groups, limiting generalisability[50]. Third, the population was almost entirely unvaccinated during the main study period. Whether the influence of mass vaccination is different for backward versus forward contact tracing remains unclear and merits further study. Fourth, the dominant variant circulating in the population during the main study period was the Alpha strain, with lower transmissibility than the subsequent Delta and Omicron VOCs (Supplementary Fig. 10)[41,42]. Our analyses of periods dominated by Delta and Omicron strains do not allow the same strong conclusions due to reduced data quality. Fifth, we did not systematically evaluate behavioural factors such as compliance with restrictions, testing, tracing, quarantine and isolation, all of which may influence the effectiveness and evaluation of tracing strategies. Sixth, a testing and contact tracing programme is a complex public health intervention, and the particular methods of implementation and contextual factors have a major impact on its overall effectiveness. The influence of host-, pathogen- and environment-related factors on the comparative efficiency of backward contact tracing strategies merits further study.

Our results indicate that in the context of significant community transmission of COVID-19 and in the presence of moderate contact restrictions, there can be a marked added benefit, at low relative cost, to extending the contact tracing window backward beyond the infectious period of the index case.

## Methods

### Study design and context

In this cohort study we investigated the risk of contracting COVID-19 for contacts traced in an extended contact tracing window. Their risk was compared to a control group of patients from the target population, who were tested for self-reported symptoms of COVID-19 in the same period (Supplementary Fig. 1).

A second reference group consisted of contacts exposed to an index case during the standard "forward" contact tracing window. The main outcome measure was a positive test in the 14 days after the last contact with the index case, or—for the control group—after the onset of symptoms.

The study was performed in the context of a dedicated test and trace system for a target population of an estimated 32,965 higher education students residing in the city of Leuven, Belgium. A low-threshold test centre offered free RT-qPCR tests upon self-referral, while a team of contact tracers performed manual bidirectional contact tracing. The programme relied heavily on community involvement and benefited from maximum integration of testing and tracing from a human process and information technology point of view. We elaborate on the operational aspects in a published testing and contact tracing protocol and show the delays involved in each step in this cascade during the study period in Supplementary Fig. 2[51].

The study protocol was approved by the Ethics Committee Research UZ/KU Leuven. Informed consent was waived as the data gathered did not exceed what was required for the purpose of safeguarding public health.

We followed the Strengthening the Reporting of Observational Studies in Epidemiology (STROBE) guidelines[52].

### Study participants

Students attached to one of Leuven's tertiary education facilities were included in the study if they either had a positive RT-qPCR test result at the KU Leuven test centre or if they were reported to the tracing team as having had a positive RT-qPCR test result elsewhere and had recently resided in or had come into contact with others in the city of Leuven. The main analysis included cases testing positive from 1st February until 31st May 2021 and their contacts.

Cases were excluded if the treating physician interpreted the result as falsely positive, or as a past infection with COVID-19. Cases who could not be contacted by the tracing team after repeated attempts were also excluded, as well as cases where information on symptom onset was missing.

Cases were asked about all their close interactions with contact persons in the period from 7 days before symptom onset or test until the time of the contact tracing interview.

Contacts were included as a close contact if they were reported by the index case as having had either direct physical contact, an interaction at less than 1.5 metres without face masks, an interaction at less than 1.5 metres for more than 15 min, or an interaction without face masks for more than 15 min. Also included as close contacts were co-attendants at a "high-risk event" of up to 20 attendees, defined as fitting at least two of the following three criteria: crowding (at least five individuals belonging to at least two households), close contact (<1.5 metres without masks) and closed environment (indoor).

Individuals who were already identified as contacts exposed to a previously diagnosed index case within 7 days before the contact tracing interview were excluded as contacts from the second identified index case, while still being considered as contacts for the first.

While this approach introduces ambiguity as to the exact day of last exposure, it is reflective of our focus on decision making at the time of first identification of a contact.

Contacts who had already tested positive on the same day as the index case or up to 60 days before, were also excluded. All other contacts were advised to quarantine while undergoing RT-qPCR testing as soon as possible and, if the test was negative, seven days after the last exposure to a positive case.

Contacts were assigned to either the standard tracing window group (a reference group mirroring standard practice) or to the extended tracing window group, based on when their last close contact with the index case took place.

As a control group, we selected all students who attended the test centre for the first time during the study period, and who self-reported symptoms suggestive for COVID-19 as the reason for their test. Only the first test was included, to reduce selection bias towards students with a lower threshold for testing.

When comparing the symptom onset date of contacts to the sampling or onset date of their index case, the analysis was restricted to case-contact pairs where the contact was also included as a case in the main analysis.

When computing the timing of testing after last exposure, the analysis was restricted to pairs where the contact was tested in the university testing centre and thus a student, as testing of other contacts didn't fall under the responsibility of the university contact tracing team and therefore was not subjected to similarly rigorous follow-up.

In the analysis assessing the sensitivity of RT-qPCR testing depending on the day after exposure, a contact was only labelled as "not infected" if they had a negative test between 7 and 14 days after last exposure. Test sensitivity on a particular day post-exposure was calculated for infected contacts who had not yet been diagnosed and was defined as the number of positive tests divided by the total number of tests in this group.

For the analyses of more recent cohorts, time periods were chosen according to the main circulating VOC and the lost to follow-up rates of contacts (Supplementary Figs. 3 and 10). All index cases and their respective contacts were included by means of the same inclusion criteria as for the main Alpha-dominant period. Inclusion and exclusion flowcharts are shown in Supplementary Fig. 6.

### Data sources
For cases and contacts tested in our test centre, RT-qPCR test results were reported directly by the laboratory. Students who tested positive elsewhere were reported by the government contact tracing teams, by the infected students themselves or by their contacts attending the test centre. The date of onset of symptoms was reported by the index case when attending the test centre and confirmed when being called by the contact tracing team.

For each of their listed close contacts, we asked the index case about the dates and nature of their interactions, and the type of their relationship. Cases could supply this information using an online web form, and were contacted by telephone for confirmation and clarification during a thorough interview. Contacts were grouped into events if multiple people were present at the same time. These contact data were coded into a customised version of Go.Data, an outbreak investigation tool developed by the WHO and Global Outbreak Alert and Response Network partners[53].

Test dates and results of contacts who were tested outside of our test centre were obtained by telephone. This information was coded into Go.Data in a similar fashion.

### Variables
Contacts were assigned one of three possible outcomes. "Infected" includes those contacts who were diagnosed with COVID-19 1 to 14 days after the diagnosis of the index case. "Not infected" denotes other contacts who underwent an RT-qPCR or antigen test with a negative result 1 to 7 days after their last contact with the index case. All other contacts were considered "lost to follow-up". In the last period, with the Omicron strain dominant, government-mandated testing for close contacts was abandoned, leading to very low testing rates[45,46]. Therefore, contacts who did not develop symptoms or undergo testing in the 7 days after exposure were considered "not infected" during this period.

The day of last contact was defined as the difference in days between the last date of interaction with the index case on the one hand, and on the other hand either the date of the positive test or the date of onset of symptoms, whichever was earlier.

Each contact of an index case was assigned a relationship type from the following list: partner, family, friend, fellow resident, acquaintance, fellow student or other.

Suspected source events were defined as events which, at the time of the contact tracing interview with the index case, were identified as the likely source of the infection, because the index case knew that an individual was present with a confirmed infection or suggestive symptoms. If the index case had been in quarantine since travelling from abroad, travel was considered the source event and travel companions were considered present. Multiple suspected source events were taken into account per index case if applicable. Suspected source events were required to fall within the backward tracing window at least partly to be labelled as such.

### Study size
The data feeding into this study were gathered in the light of the ongoing public health response for COVID-19. The exact study period was chosen from February onwards since gradual improvements in data gathering—through updates of the IT infrastructure and human capacity building—allowed for follow-ups of all contacts to be consistently recorded from February onwards. The end of the main study period marks the end of the academic semester, at which point testing and case numbers fell precipitously. The resulting number of cases and contacts is a consequence of the epidemiological trajectory within the study period.

### Statistical methods
Data analysis was performed either using R script in R version 4.0 .3 or python script in python version 3.8, specifically written for this study.

PR were calculated with two-sided 95% confidence intervals according to the Clopper–Pearson method. Small-sample adjusted risk ratios were determined with two-sided normal approximation 95% confidence intervals.

Missing demographic data was ignored in the calculations and the amount of missing data reported. Contacts with missing outcome data were considered lost to follow-up.

Cases and contacts lost to follow-up were not included in the analysis.

### Reporting summary
Further information on research design is available in the Nature Research Reporting Summary linked to this article.

## Data availability
The data underlying the main analyses in this manuscript are available in the article and in its online Supplementary Material. The data that is not released with the paper, and which may require EC approval before sharing, can be made available on request from the corresponding author (J.R.), who will respond within 4 weeks. There must be a demonstrable affiliation with an academic or health institution, a legitimate epidemiological question and a commitment to not attempt to de-anonymise. Source data are provided with this paper.

## Code availability

The code of the iterative contact tracing model is available in Supplementary Data 1.

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

## Acknowledgements

We thank everyone involved in the set-up and running of the KU Leuven test and trace programme, especially the many student workers who formed the bulk of the team and Prof Dr Jan Verhaegen and Prof Dr Paul De Cock for their many hours volunteering. We also thank the senior supervisory team consisting of Prof Dr Chris Van Geet, Prof Jef Arnout, Mr Bruno Lambrecht, Prof Leen Delang, Dr Rikka De Roy, Dr Anja Vandeputte, Prof Dr Katrien Lagrou and Mr Lieven Put as well as Dr Femke Kerkhofs and Mr Cis Dejonckheere for their work on coordinating the testing centre and contact tracing operations. The test centre was funded by both the National Institute for Health and Disability Insurance (RIZIV/INAMI) and the regional Flemish government's Agentschap voor Zorg & Gezondheid. The contact tracing programme benefited from the generous contributions of KU Leuven University. J.R. is funded through an FWO strategic basic research fellowship (1S88721N). Data on student numbers was provided by Mr Kris Cuppens and Mr Jeroen De Keyser from the KU Leuven and KU Leuven association data cells. Sciensano provided data on test results at country level. We thank Mr Ruben Brondeel and Mr Dieter Van Cauteren for their help in retrieving the data.

## Author contributions

J.R., K.N., C.G. and E.A. conceived and designed the analysis. C.G., J.R., K.N. and S.G. collected the data. S.G. contributed data. C.G., J.R. and J.T. performed the analysis. J.R. and C.G. wrote the paper. K.N., J.T., S.G. and E.A. critically reviewed the paper.

## Competing interests

The authors declare no competing interests.
