## [Peer Review File · Nature Communications]

Empirical evidence on the efficiency of backward contact tracing in COVID-19Editorial Note: Parts of this Peer Review File have been redacted as indicated to maintain the confidentiality of unpublished data.

REVIEWER COMMENTS

Reviewer #1 (Remarks to the Author):

Review of NCOMMS-21-40303-T

Title: Empirical evidence on the efficiency of bidirectional contact tracing in COVID-19

Reviewer: Tim Lucas, University of Leicester

Date: 2021-11-04

Summary

In this study the authors aim to test the effectiveness of bidirectional contact tracing. The study was conducted in a higher education establishment. Given that contact has been relatively ineffective in much of Europe during the pandemic, methods to make it more effective are very welcome. Modelling studies have claimed that bidirectional tracing can greatly improve contact tracing efficiency (though I am quite critical of at least one of the referenced modelling studies).

The study was conducted by tracing individuals in a seven day window preceding symptoms/positive test. Any contacts who were in close-contact with the index case between 7 and 3 days preceding were considered to be part of reverse contact tracing, while cases who were in close-contact 1 or 2 days preceding were considered to be part of forward contact tracing. The main results are that the positivity of contacts in the reverse contact tracing group was similar to those in the forward contact tracing group and that both of these groups had considerably higher positivity than a reference group of symptomatic but un-traced individuals. The authors interpret this as evidence that reverse contact tracing works.

I think this is an important study with important data. I'm therefore glad to see that the authors have preprinted the work. However, I think there are some problems with how the data are interpreted given the experimental setup as described below. I also think a number of aspects of the methods in particular could be clearer.

Major issues

The main issue with the study is that it does not test the full process of bidirectional contact tracing. It tests the ability of the contact tracers to identify cases (which are presumed to be either parent or child cases respectively). However, this is not the specific goal of contact tracing. The goal of contact tracing is to identify cases and isolate them before they transmit the pathogen.

Reverse contact tracing is difficult because to prevent onwards infections, either the parent case has to be identified while still infectious or (more importantly) sibling cases (i.e. other contacts of the parent case) must be isolated before they become infectious. In general, given a window of 7-days, by the time the parent contact has been found and is in isolation, half or more of their potential child infections would have already occurred (pane c here

<https://www.nature.com/articles/s41591-020-0869-5/figures/1>).

Therefore, the main potential benefit of reverse contact tracing comes from being able to identify sibling cases before they become infectious.

Given the short serial interval of SARS-CoV-2, the delay caused by the gap between infection and symptom onset, and the typical delays in contact tracing (on the order of days) this is difficult to achieve (in contrast to diseases such as STIs where I believe reverse contact tracing was originally developed).

While this still may be possible, it is unfortunate that this full process of reverse contact tracing is not tested in this analysis.

It might be possible that the authors are able to measure this from the data they already have. If they can identify (in their data) the contacts of the cases that were from the 3-7 window they could compare the positivity of these people to the positivity of the control group and the 1-2 day window individuals.

If they cannot retrieve this information from their data (due to a lack of linkage) I personally think it is still useful to publish the study and data as is, but I think large caveats and changes in conclusions must be made.

For example I don't believe that "there is a marked added benefit" (line 275) and "imply an urgent need to implement backward contact tracing globally" (line 30) are supported by this study.

In my opinion the study suggests that reverse contact tracing might be useful, if other hurdles such as delays times, can be overcome.

Minor issues

1) I spent quite a long time totally misunderstanding the design of the study.

I originally thought that the comparison was forward tracing only (days 1-2 window) versus bidirectional tracing (days 3-7 window).

Alongside this I thought the hypothesis was that bidirectional tracing is better than forward tracing (prevalence in bidirectional tracing group is higher than prevalence in forward tracing group).

And the design doesn't make sense for this hypothesis and the results don't support the hypothesis.

Only after careful repeated reading did I realise that the design was forward tracing only (days 1-2 window) versus backward tracing only (3-7 window) and the hypothesis was that backwards is of similar effectiveness as forward tracing (which to give credit is stated explicitly in line 123).

There is then a second layer that bidirectional tracing (days 1-2 AND 3-7) is better than forward only tracing (days 1-2).

I think making this whole overview very explicit in both the intro and the methods would be useful (I doubt I'm the only one who reads the abstract then skips to the methods).

Perhaps adding this in the results or discussion as well would be useful given Nature's insistence on putting the methods at the end.

2) There is no mention of isolating contacts who have not been infected. This is one of the costs of contact tracing and probably should be mentioned. This does however depend on whether the contact tracing system asks (or demands) that people isolate immediately once traced (as in the UK) or whether they are asked to take a test once traced.

3) Figure 1 is very useful for giving an overview of how backwards contact tracing works. However, it doesn't convey the temporal aspects i.e. that you need to identify the parent contact and the subsequently identify the sibling contacts in time to isolate them before they become infectious. I think a sentence or two, or an additional figure, highlighting these a priori difficulties in backwards contact tracing in the context of SARS-CoV-2 would be useful.

In this same section it might be useful to note that the windows (1-2 days vs 3-7 days) do not

guarantee that a contact is a child versus a parent case due to various overlaps (this is one of the problems with modelling reverse contact tracing using branching processes for example).

4) The control group is not quite the natural control group which would probably be random background testing or something instead.

This is fine and I understand why the design is like that, but I think it might be worth adding a sentence mentioning that this is sort of a rough approximation to a control group or something. I actually don't even know what the true control group would be even without any operational constraints.

5) It is slightly surprising that fewer contacts are identified in the extended window (5 days long, from day 3 to 7) than in the standard window (2 days long) as shown in figure 2.

I imagine this is because of cases forgetting who they were in contact with.

You mention this in passing in the discussion but I think it should be discussed explicitly.

It is not clear whether it is a benefit (honing in on the most important contacts) or a flaw (higher chance of missing the true parent case).

It also relates to my previous comment on falsely isolating contacts who have not been infected.

Reviewer #2 (Remarks to the Author):

Contact tracing is one of the three key strategies that health authorities deployed to slow and hinder the spread of Covid-19. So far, forward contact tracing has been the main mode of operation. Forward contact tracing identifies individuals that a confirmed index case has been in contact with during the infectious period. Hence, it makes no attempts to identify who may have infected the index case. There is, however, a growing evidence from theoretical studies that augmenting current practices by extending the tracing window beyond the infectious period (i.e. backward contact tracing), can reveal more cases and consequently limit the spread of the virus.

This paper presents a large cohort study on backward contact tracing. More specifically, it tracked infections among students in the city of Leuven, Belgium. It measured the risk of contracting Covid-19 for contacts identified via backward contact tracing using a window of seven days prior to symptom onset. Then compared that to the same risk for individual that were identified using forward contact tracing as well as to a control group from the target population.

The results of the study confirm the previously published theoretical insights. Using backward contact tracing increased the number of identified contacts by 42%. Further, the benefit was not limited to flagging attendees of crowded social events.

Comments

I found the results interesting and encouraging. However, I would have liked to see more analysis of the factors that can impact the effectiveness of backward contact tracing. For example, It is known that delays in testing can reduce the effectiveness of contact tracing and I expect this to have a much bigger impact on backward contact tracing. I recommend the authors to look into the impact of timely testing on the number of positive contacts that backward contact tracing was able to identify. I also wonder whether the authors have considered including faster methods of testing like antigen tests?

A key benefit from backward contact tracing is the fact that it helps identifying the source that infected the index case in question. This in turn can help isolating micro outbreaks. Apart from looking at the crowded events, the paper did almost nothing to infer chains of transmission. Inferring these chains can help understanding the role of timely testing as well as identifying potential superspreaders.

Backward contact tracing can result in recommending more individuals to quarantine than forward contact tracing, which may have severe social and economical impact. At the same time, one can

argue that backward contact tracing can lead to less quarantining overall. Again, the paper could have tried to explore this a bit more.

The study excluded contacts (324 contacts) who were already exposed to another diagnosed index case in the seven days leading to the contact tracing interview. This is understandable from a contact tracing point of view, but I wonder whether these cases were in the end somehow related (i.e. part of the same micro-outbreak)?

I liked the analysis of source events, however, defining whether a person was infected because of a particular source event needs more sharpening. This question is essentially a timeline question. Whether an infection can be causally linked to a crowded event depends on whether the index was infected at the event and on when the index did pass on the infection.

The high positivity rate is quite interesting and I would have liked to see more analysis that can help shed light on the underlying mechanisms. Potential explanations as the authors mentioned is common sources of infection, index cases that were contagious before the symptoms onset, etc. The dataset can probably help isolating some of these causes by, for example, examining differences between individual cases as well as transmission chains.

Reviewer #3 (Remarks to the Author):

Raymenants et al collected and analysed SARS-CoV-2 contact tracing data with a view to evaluating the potential for improvements through 'backward' contact tracing: identifying the person who infected an index case, rather than the more usual identification of people the index case infected ('forward' tracing). The data gathered is appropriate, albeit in a subpopulation of higher-education students with limited generalisability, as the authors already concede. The analyses are simple but appropriate, essentially calculating fractions within different subsets of the data. The writing is clear and accessible.

Journal guidance to reviewers asks "do you feel that the results presented are of immediate interest to many people in your own discipline and/or to people from several disciplines?" It also states that "a paper should represent an advance in understanding likely to influence thinking in the field, with strong evidence for their conclusions. There should be a discernible reason why the work deserves the visibility of publication in a Nature Portfolio journal rather than the best of the specialist journals." I think that these results will only be of interest to people focussed on COVID-19 contact tracing, and novelty has been overstated, and I haven't identified a discernible reason for publication outside of the best of the specialist journals. On the other hand, the results are useful, and the topic of COVID-19 contact tracing is currently important for public health, and I agree with the authors that starting the tracing window at least a little bit earlier and investigating source events are both warranted. I therefore defer more than usual to the editor's discretion on acceptance.

The introduction focuses on backward contact tracing as a concept distinct from forward tracing, as supported by previous theoretical work, and states that no study until now has evaluated this approach. I think this framing overstates the novelty of the current study. An alternative lens through which these results can be considered is as follows. The authors identified contacts linked to index cases, calculated the positivity rate among these individuals (the fraction of them testing positive, usually called the secondary attack rate in such studies) when disaggregating by different characteristics or covariates, in order to see which of these are associated with greater risk of transmission. This is a study design that has been widely used and reported throughout the pandemic, for example Koh et al published a meta-analysis of 57 such studies in 2020 <https://journals.plos.org/plosone/article?id=10.1371/journal.pone.0240205>. As the authors note, backward contact tracing in practise is either through (A) extending the tracing window further back in time than the usual 2 days before the case's symptoms or diagnosis, and/or (B) explicitly attempting to determine the event at which the case became infected - "source investigation". Policy decisions on the two actionable points A and B are informed by the two main quantities

measured here: (a) how likely contacts are to test positive as a function of when they were exposed relative to the case's symptoms or diagnosis, and (b) how likely contacts are to test positive depending on whether they were at the suspected source event. Previous studies measuring (a), even if they did not discuss the possibility of a sibling rather than a parent-child relationship for transmission, include

<https://www.nature.com/articles/s41591-020-0869-5>

<https://jamanetwork.com/journals/jamainternalmedicine/fullarticle/2765641>

<https://www.medrxiv.org/content/10.1101/2020.09.04.20188516v2>

The authors note that source investigation is already used by several countries, citing four references in support of it, to which could be added its recommendation by WHO

https://apps.who.int/iris/bitstream/handle/10665/339128/WHO-2019-nCoV-Contact_Tracing-2021.1-eng.pdf?sequence=24&isAllowed=y and the US CDC

<https://www.cdc.gov/coronavirus/2019-ncov/php/contact-tracing/contact-tracing-plan/source-investigation.html>. The results here could therefore be alternatively framed as further measurement of the relationship between risk and time of exposure, and further evidence in support of source investigation, which would be less striking but more fair than the first-study-of-its-kind claim made currently.

Major suggestions

The paper's conclusion as phrased in the abstract - "Our results imply an urgent need to implement backward contact tracing globally" - is by some way too strong based on the evidence presented here. One of five limitations acknowledged by the authors is

265 "index cases were young adults in tertiary education, whose socio-economic status
and contact patterns may differ significantly from other age and social groups, limiting
generalisability"

This study provides sufficient motivation not for immediate implementation of policy globally, but for further pilot studies to be conducted by contact tracing teams operating at different hierarchies - local, regional, national - with more representative populations and in different geographic, socio-economic and epidemic contexts. By testing the extra contacts identified by this procedure, without imposing a requirement to quarantine, the main cost of the intervention (additional unnecessary quarantine) can be avoided while gathering more evidence for evaluation. One would therefore hope to find a good appetite for embedding such pilot studies in existing tracing programmes.

The paper's advocacy of backward tracing is based only on the number or fraction of contacts identified who are infected. The extent to which contact tracing reduces transmission from each of these individuals depends on two factors: how much they reduce their infectiousness after they are traced (e.g. by isolation) and when they were traced relative to their infectious period (i.e. tracing them a fraction x of the way through their infectious period means a fraction $1-x$ is left to be affected by the intervention). The first point is sufficiently obvious as to not need special mention here. The second point is critical even for purely forward tracing, due to the short generation time and pre-symptomatic infectiousness of SARS-CoV-2 infections. It is more important still for backward tracing: going backwards in the chain of transmission means infected contacts identified this way will be identified later in their infectious period. If they are typically identified toward the end of their infectious period, tracing them will have little effect. Bidirectional iterative tracing will mitigate this to some extent, but I expect the average effect will always be a later identification of contacts than in purely forward tracing. The authors' data could quantify this effect in two ways. Firstly, by comparing the time at which contacts were traced to the time of their onset of symptoms, as a function of when the contact was exposed relative to the index case's symptoms. The general trend should be that as we move earlier into the extended tracing window towards -7 days, the later the tracing will be relative to the contact's symptoms (and by implication, relative to their infectious period). Secondly, by comparing the time at which contacts were traced to the time of the suspected source event, for those contacts who also attended. If neither of these time intervals can be obtained from the data, the general point concerning timings should be acknowledged as a caveat for this evaluation and for backward tracing generally.

503 "The data that support the findings of this study are available on request from the
504 corresponding author (Joren Raymenants). The data are not publicly available because they

505 could compromise research participant privacy.”

The raw data at point of collection is of course sensitive and should not be public. However, after it has been processed and simplified to the point of entry in the main analysis, it consists of only the following variables for each traced contact:

- was their outcome: a positive test, a negative test, or lost to follow up;
- how many days were in between their last exposure to the index case and the index case's onset of symptoms or diagnosis (note: neither of the two dates is required, only their difference);
- was the contact at the case's suspected source event: yes, no, or source event not identified;
- what kind of relationship is there between case and contact: friend, family etc.

These data do not permit identification of the individuals in question, and would allow reproduction of the authors' main results from the data (and possible extensions, for example fitting a smooth parametric function to the transmission risk over subsequent days). Inclusion of the data in this form as supplementary material is recommended. Additionally, it was reported that “72.5% of index cases self-reported being symptomatic at the time of testing”. Simply adding a binary variable for each contact noting whether they were associated with these cases would provide partial discrimination between the cases' symptomatic infectious phases and their pre-symptomatic/asymptomatic phases. Additionally noting when the index did develop symptoms relative to their diagnoses (again, without actual dates, just differences) would allow full discrimination of these phases. Quantifying the difference in transmission risk between these phases is an important epidemiological issue extending beyond contact tracing.

For data that is not released with the paper, the criteria that those wishing to access it must satisfy should be stated here. For example a demonstrable affiliation with an academic or health institution, a legitimate epidemiological question, commitment to not attempt to de-anonymise etc.

Contact tracing is a cost-benefit trade-off. The authors acknowledge this to some extent, implicitly, by basing their recommended expansion of the tracing window upon the observation that it has minimal effect on the efficiency of identifying infected individuals (i.e. similar positivity rates in the two traced groups). Some explicit acknowledgment of the trade-off would be welcome, both in the introduction and the discussion. For example in the introduction, one paragraph discusses what previous modelling studies say about backward tracing, but only with regards to its benefits. Did these studies comment on the costs? For example the fraction of backward traced individuals who are not actually infected, or how much extra quarantine is required for the epidemiological gains? Did these studies address the previous paragraph's point concerning speed? A cost to be acknowledged in the discussion is that, even if backward tracing identified infected individuals with exactly the same efficiency as forward tracing, tracing more individuals requires more work if done manually. Expanding tracing therefore requires hiring additional contact tracers, and/or reduces the maximum number of index cases manual tracing can handle.

Minor suggestions

The authors could comment on possible differences in implementation between digital and manual contact tracing. For example with digital tracing, expanding the window of exposure would only require changing the value of a parameter of the algorithm, not an expansion in the tracing team's workload; however, interviewing the case about their suspected source event is not possible. The NHS COVID-19 App in England and Wales sends 'risky venue' warning notifications to those who anonymously 'checked in' with the app to a venue that later becomes linked to enough cases.

The point of reference for comparison of positivity rates is individuals who have self-referred for testing based on symptoms. It should be acknowledged that the positivity rate in such a control group could vary widely depending on the incidence of SARS-CoV-2 infections in that place, time and subpopulation, and the incidence of other respiratory illnesses at that place, time and subpopulation.

The authors refer to "an incubation period" / "the incubation period" without acknowledging variability in this quantity between individuals, as captured by the incubation period distribution. Since backward tracing is based on going back in time by one incubation period, at least a short discussion of this distribution is warranted. The greater the variance of the distribution, the harder it is to estimate when an individual was infected.

The authors already note that the impact of backward contact tracing is linked to overdispersion in the offspring distribution (the number of individuals to which each infected individual transmits). Since transmission risk can be decomposed into the product of biological infectiousness (the rate of shedding viable virions) and the rate & intensity of contact events with susceptible individuals, the overdispersion in actual transmission events can be decomposed into contributions from overdispersion in biological infectiousness (with cross-sectional viral load surveys being one proxy) and overdispersion in social contact networks. Where either of these factors is expected to be variable depending on the subpopulation - and presumably the second factor at least is - this provides a specific example of the stated limitation about the representativeness of higher education students as a study group. Which could be commented on, if desired.

"Case-based interventions such as case isolation, contact tracing and quarantine"
Quarantine is not always a case-based intervention: it is for those at risk of being infectious for some reason, for example travel. I suggest "such as case isolation or contact tracing with quarantine" instead.

"In addition to child cases, any practical
forward tracing strategy probably identifies the parent case (the infector of the index case)
and sibling cases (infected by the same parent case) some of the time"
It would be good to clarify here why this happens, because the frequency with which it happens determines the additional benefit of backward-by-design tracing. i.e. either the case has a second contact with their infector during their own infectious period, or the time from the case's infection to their symptoms or diagnosis was less than two days.

The authors mention that tracing iteratively can improve reach, in the sentence after backward tracing is first introduced. I suggest acknowledging very briefly that iterative tracing extends the reach of purely forward tracing too, lest it be mistaken as something specific to backward or bidirectional tracing.

"High
positivity rates are reported for attendants of such source events"
The phrasing sounds like positivity is high at all events with suspected transmission, but sometimes the suspicion will be wrong. I suggest "have been reported... at some source events"

91 "This approach [source investigation] is reliant on
the identification of multiple infected cases at the same event, by pooling of contact tracing
data from different index cases"
I assume the authors mean in practice but not in theory, because even just a single case can speculate on which event they were infected at, and if I have understood the results presented here that is precisely what happened. If the statement above applies to practice, its scope should be clarified and supporting evidence cited. I suspect that there will be many contexts where this is not true - where contact tracing teams investigate events where a single case might have been infected, without (or before) a second case being linked to that event.

"both as soon as possible and after an
incubation period."
Confusing phrasing - the two aspects seem to contradict each other.

"Fyles et al also show in a branching process model that an extended contact tracing window
results in a linear decrease in the growth rate up until around 8 to 10 days prior to symptom
onset or diagnosis[11]."

An important caveat from reference [11] should be acknowledged:
"However, identified cases might struggle to remember contacts they have had back in the past. We find that including a daily 10% reduction in the probability of recalling a contact in the model erodes the gains of backwards contact tracing almost completely"

"identified in an extended contact tracing window, starting 7 days before onset of symptoms
or

119 diagnosis."

I suggest adding here "(whichever was earlier)"

"at the time of writing"

I suggest replacing by a specific moment in time for ease of reference in future.

163 "We did not collect demographic data on contacts of index cases..."

There was a slightly higher percentage of women in the control group (56.5%, missing data
3.0%) while the mean age was similar to the other groups"

To which groups is the control group being compared to here? I interpreted line 163 as meaning demographic data is unavailable for the standard tracing window group and the study group.

213 "backward contact tracing..."

214 identifies an additional 43.5%

215 of cases"

Use of the present tense suggests this is a universal truth. Past tense, and restriction of the sentence to the present study, would clarify the scope.

"We excluded contacts already identified as exposed within 7 days before the contact tracing
interview to a previously diagnosed index case."

I think this means that such contacts were excluded from the study entirely; however, it would be good to clarify whether they were only excluded as contacts from the second identified index case, while still being considered as contacts for the first. In that scenario, the contact's exposure time is ambiguous without further details.

A comma is used as a decimal separator in the results figure, whereas a point is used in the prose.

Extended data Figure 1 needs an x axis label.

No comments have been provided to the editor separately from those here.

With best wishes to the authors,
Chris Wymant

Dear Editor,

We would like to thank each of the reviewers for the time and effort they have taken to provide feedback relating to our manuscript. We are pleased to provide a manuscript with major revisions, incorporating most of the suggested changes, including additional analyses regarding the timeline aspects of backward contact tracing. The main analysis has been repeated for data gathered during more recent periods, even if conclusions from these are weakened by low follow-up rates. We have highlighted the changes in the manuscript.

The following is a point-by-point response to each of the reviewers' comments.

Reviewer #1 (Remarks to the Author):

Review of NCOMMS-21-40303-T

Title: Empirical evidence on the efficiency of backward contact tracing in COVID-19

Date: 2021-11-04

Summary

In this study the authors aim to test the effectiveness of bidirectional contact tracing. The study was conducted in a higher education establishment. Given that contact has been relatively ineffective in much of Europe during the pandemic, methods to make it more effective are very welcome. Modelling studies have claimed that bidirectional tracing can greatly improve contact tracing efficiency (though I am quite critical of at least one of the referenced modelling studies).

The study was conducted by tracing individuals in a seven day window preceding symptoms/positive test.

Any contacts who were in close-contact with the index case between 7 and 3 days preceding were considered to be part of reverse contact tracing, while cases who were in close-contact 1 or 2 days preceding were considered to be part of forward contact tracing.

The main results are that the positivity of contacts in the reverse contact tracing group was similar to those in the forward contact tracing group and that both of these groups had considerably higher positivity than a reference group of symptomatic but un-traced individuals.

The authors interpret this as evidence that reverse contact tracing works.

I think this is an important study with important data.

I'm therefore glad to see that the authors have preprinted the work.

However, I think there are some problems with how the data are interpreted given the experimental setup as described below.

I also think a number of aspects of the methods in particular could be clearer.

Major issues

The main issue with the study is that it does not test the full process of bidirectional contact tracing. It tests the ability of the contact tracers to identify cases (which are presumed to be either parent of child cases respectively). However, this is not the specific goal of contact tracing. The goal of contact tracing is to identify cases and isolate them before they transmit the pathogen.

Reverse contact tracing is difficult because to prevent onwards infections, either the parent case has to be identified while still infectious or (more importantly) sibling cases (i.e. other contacts of the parent case) must be isolated before they become infectious.

In general, given a window of 7-days, by the time the parent contact has been found and is in isolation, half or more of their potential child infections would have already occurred (pane c here <https://www.nature.com/articles/s41591-020-0869-5/figures/1>).

Therefore, the main potential benefit of reverse contact tracing comes from being able to identify sibling cases before they become infectious.

Given the short serial interval of SARS-CoV-2, the delay caused by the gap between infection and symptom onset, and the typical delays in contact tracing (on the order of days) this is difficult to achieve (in contrast to diseases such as STIs where I believe reverse contact tracing was originally developed).

While this still may be possible, it is unfortunate that this full process of reverse contact tracing is not tested in this analysis.

It might be possible that the authors are able to measure this from the data they already have. If they can identify (in their data) the contacts of the cases that were from the 3-7 window they could compare the positivity of these people to the positivity of the control group and the 1-2 day window individuals.

If they cannot retrieve this information from their data (due to a lack of linkage) I personally think it is still useful to publish the study and data as is, but I think large caveats and changes in conclusions must be made.

For example I don't believe that "there is a marked added benefit" (line 275) and "imply an urgent need to implement backward contact tracing globally" (line 30) are supported by this study.

In my opinion **the study suggests that reverse contact tracing might be useful, if other hurdles such as delays times, can be overcome.**

We agree that backward contact tracing risks isolating infected contacts later in their infectious period, which reduces its effect on onward transmission. We also agree that its effectiveness hinges on the isolation of infected contacts before their infectiousness has subsided and that rapid detection of sibling cases is therefore paramount. We disagree however that the only way to identify sibling cases is through identification of the parent case and their secondary contacts, which is indeed how backward contact tracing is generally conceived in modelling studies. In further analysis of our data, added in the revised manuscript, the relative timings of symptom onset of index cases and their infected contacts implies that most backward traced contacts are sibling cases exposed to the same (unidentified) source (*Figure 5, panel a*) and can thus be identified directly through contact tracing of the index case. We opted for this analysis since we feared that the analysis proposed by the reviewer would be influenced by the complex dynamics taking place in the social network of cases and their contacts. We computed the delay period between the testing of an index case on the one hand and the onset of symptoms of their infected contacts on the other. When stratifying contacts to the forward or backward tracing window, we showed that backward traced contacts – which were exposed on average 4 days earlier than their forward traced counterparts – are quarantined on average 1.8 days later in their infectious cycle (see “Backward contact tracing identifies mostly sibling cases” and *Figure 5, panel b*). This delay, which could be interpreted as a reduction in contact tracing efficiency equal to an additional testing or tracing delay of the same period, likely does have an influence on the overall effectiveness of backward contact tracing. The lower testing and quarantine

burden placed on these contacts does however improve the cost-benefit ratio of the approach (see “Less tests and shorter quarantine in the backward traced group”, *Figure 6* and *Supplementary Figure 5*).

We have nuanced both statements highlighted as follows:

"there is a marked added benefit" (line 275) has become: “Our results indicate that in the context of significant community transmission of COVID-19 and in the presence of moderate contact restrictions, there can be a marked added benefit, at low relative cost, to extending the contact tracing window backward beyond the infectious period of the index case.”

"imply an urgent need to implement backward contact tracing globally" (line 30) has become “Our results support implementing backward contact tracing when rigorous suppression of viral transmission is warranted.”

Minor issues

1) I spent quite a long time totally misunderstanding the design of the study.

I originally thought that the comparison was forward tracing only (days 1-2 window) versus bidirectional tracing (days 3-7 window).

Alongside this I thought the hypothesis was that bidirectional tracing is better than forward tracing (prevalence in bidirectional tracing group is higher than prevalence in forward tracing group).

And the design doesn't make sense for this hypothesis and the results don't support the hypothesis.

Only after careful repeated reading did I realise that the design was forward tracing only (days 1-2 window) versus backward tracing only (3-7 window) and the hypothesis was that backwards is of similar effectiveness as forward tracing (which to give credit is stated explicitly in line 123).

There is then a second layer that bidirectional tracing (days 1-2 AND 3-7) is better than forward only tracing (days 1-2).

I think making this whole overview very explicit in both the intro and the methods would be useful (I doubt I'm the only one who reads the abstract then skips to the methods).

Perhaps adding this in the results or discussion as well would be useful given Natures' insistence on putting the methods at the end.

We thank you for your comment. We changed the title of the manuscript to stress that we assessed the added benefit of backward contact tracing – through the implementation of an extended contact tracing window – in comparison to a standard contact tracing window. The different approaches were made more explicit in the introduction and *Figure 1*.

2) There is no mention of isolating contacts who have not been infected. This is one of the costs of contact tracing and probably should be mentioned. This does however depend on whether the contact tracing system asks (or demands) that people isolate immediately once traced (as in the UK) or whether they are asked to take a test once traced.

Although we did mention the number of additional contacts identified in the backward contact tracing window (n=793), the relative increase they represented as opposed to those in the standard tracing window (49%) and the relative increase in number of detected cases they represented (42%), our additional analyses in this revised manuscript looks into the risk of identifying contacts late in their infectious cycle (see above). We also demonstrate that the inherent differences in exposure dates compared to forward tracing allow to reduce the number of tests and lead to shorten durations of quarantine in backward traced contacts.

3) *Figure 1* is very useful for giving an overview of how backwards contact tracing works. However, it doesn't convey the temporal aspects i.e. that you need to identify the parent contact and the subsequently identify the sibling contacts in time to isolate them before they become infectious.

I think a sentence or two, or an additional figure, highlighting these a priori difficulties in backwards contact tracing in the context of SARS-CoV-2 would be useful.

In this same section it might be useful to note that the windows (1-2 days vs 3-7 days) do not guarantee that a contact is a child versus a parent case due to various overlaps (this is one of the problems with modelling reverse contact tracing using branching processes for example).

We added a figure outlining the different contact tracing strategies and the temporal aspects involved (Figure 1). The fact that we make no assumptions on directionality has been stressed in the caption of Figure 1 and the results section.

4) The control group is not quite the natural control group which would probably be random background testing or something instead.

This is fine and I understand why the design is like that, but I think it might be worth adding a sentence mentioning that this is sort of a rough approximation to a control group or something.

I actually don't even know what the true control group would be even without any operational constraints.

We acknowledge the limitations of the control group and have further stressed these in the discussion section: "Positivity rates amongst symptomatic individuals are dependent on many factors, such as the level of community transmission of SARS-CoV-2 and other respiratory viruses. Still, this group was chosen as a control group, because it represents a high bar and testing of symptomatic patients is standard in most protocols globally^{37,45,46}." and "The control group probably also suffered a further reduction in reliability after the main study period, due to the rollout of alternative testing methods such as pharmacy-based and self-administered rapid antigen tests and the progressive scaling back of RT-qPCR testing in general^{37,45,46}."

5) It is slightly surprising that fewer contacts are identified in the extended window (5 days long, from day 3 to 7) than in the standard window (2 days long) as shown in figure 2.

I imagine this is because of cases forgetting who they were in contact with.

You mention this in passing in the discussion but I think it should be discussed explicitly.

It is not clear whether it is a benefit (honing in on the most important contacts) or a flaw (higher chance of missing the true parent case).

It also relates to my previous comment on falsely isolating contacts who have not been infected.

This is now discussed more explicitly in the results section. In our opinion, the main reasons that fewer additional contacts are found in the backward tracing window are recall decay and recurring contacts with the same individuals. The most obvious example of the latter are household contacts. If a contact person was encountered in both the backward and the forward contact tracing window, we considered them to belong to the forward traced group only. This has been clarified in the discussion section.

Reviewer #2 (Remarks to the Author):

Contact tracing is one of the three key strategies that health authorities deployed to slow and hinder the spread of Covid-19. So far, forward contact tracing has been the main mode of operation. Forward contact tracing identifies individuals that a confirmed index case has been in contact with during the infectious period. Hence, it makes no attempts to identify who may have infected the index case. There is, however, a growing evidence from theoretical studies that augmenting current practices by extending the tracing window beyond the infectious period (i.e. backward contact tracing), can reveal more cases and consequently limit the spread of the virus.

This paper presents a large cohort study on backward contact tracing. More specifically, it tracked infections among students in the city of Leuven, Belgium. It measured the risk of contracting Covid-19 for contacts identified via backward contact tracing using a window of seven days prior to symptom onset. Then compared that to the same risk for individual that were identified using forward contact tracing as well as to a control group from the target population.

The results of the study confirm the previously published theoretical insights. Using backward contact tracing increased the number of identified contacts by 42%. Further, the benefit was not limited to flagging attendees of crowded social events.

Comments

I found the results interesting and encouraging. However, I would have liked to see more analysis of the factors that can impact the effectiveness of backward contact tracing. For example, It is known that delays in testing can reduce the effectiveness of contact tracing and I expect this to have a much bigger impact on backward contact tracing. I recommend the authors to look into the impact of timely testing on the number of positive contacts that backward contact tracing was able to identify. I also wonder whether the authors have considered including faster methods of testing like antigen tests? We agree that the speed of each step in the testing and contact tracing process is key to determining overall effectiveness, along with comprehensiveness of both testing and contact tracing. We have made several additions in the manuscript relating to these aspects.

The different delay times involved in the Test Trace Isolate Quarantine (TTIQ cascade) during the main study period are quantified in a new *Supplementary Figure 2*.

The general issue concerning timeliness of testing and contact tracing is introduced in the introduction: “One potential difficulty of backward contact tracing lies in the inherent delays involved in testing, tracing and quarantine – where infected contacts who are sibling or parent cases risk being detected after or near to the end of their infectious period^{3,18}. This could reduce efficiency and increase the relative cost of testing and quarantine (*Figure 1*). Due to these delays, immediate testing of identified contacts in support of iterative tracing may be especially relevant in backward contact tracing.”

We also agree that a potential risk of extending the contact tracing window backward lies in the fact that the infected contacts who are identified in addition may only be detected in a late stage of their infection, which would have little effect on onward spread but does raise the cost of testing and quarantine. The risk of quarantining backward traced contacts later in their infectious cycle is quantified under a new section entitled: “Backward contact tracing identifies mostly sibling cases”. In this section, we computed the delay in symptom onset between an index case and their backward and forward traced contacts (*Figure 5, panel a*) and showed that the former identifies only a minority of likely parent cases but mostly identifies sibling cases directly. The latter on the other hand identifies both sibling and child cases. In addition, we computed the delay time between testing of an index case

and symptom onset in their contacts. This analysis shows that backward traced contacts who are both infected and symptomatic develop symptoms on average 0.04 days after their respective index case is tested, while forward traced infected contacts develop symptoms after an average of 1.81 days. The difference of 1.77 days between both is significant. Infected backward traced contacts are thus detected later in their infectious cycle.

A third aspect we have had a deeper look into is the value of testing at different intervals after last exposure. Thorough contact tracing implicitly requires contacts to be tested immediately after identification, since this allows for iterative tracing of newly identified cases. In addition to adding this more explicitly to the introduction and a new *figure 1*, we also assessed the diagnostic accuracy of RT-qPCR testing depending on the number of days since last exposure to an index case. The results are added to a new *Figure 6*. It demonstrates that the sensitivity of RT-qPCR tests increases rapidly in the first days after last exposure. As backward traced contacts were detected on average 4 days later after last exposure compared to forward traced contacts, a single test could serve both a “test to trace” and a “test to release” function more often in backward traced contacts (*Figure 6 panels a and b*). *Supplementary Figure 5* demonstrates the influence of two factors on the percentage of forward and backward traced contacts requiring only a single test for both “test to trace” and “test to release”: the delay between the testing of an index and a “test to trace” in their contacts and the policy set minimal delay required between last exposure and a “test to release”. It shows that backward traced contacts can – due to their inherently different exposure dates – more often be released from quarantine after a single test, reducing both their testing and quarantine burden.

Concerning the matter of antigen testing: antigen tests were not yet widely available to the public in Belgium at the time of the study. Only 0.21% of our attendants between February and June 2021 self-referred for confirmatory testing following self-diagnosis by means of an antigen test. Another 1.02% of attendants between the 22nd of February and the 31st of May were tested by means of an NP AG test in the context of a clinical trial running at the test centre. These results are not added to the main manuscript. They are added to a published protocol (<https://doi.org/10.21203/rs.3.pex-1666/v2>). The rollout of alternative testing methods such as pharmacy-based and self-administered rapid antigen tests increased from the summer of 2021 onwards^{35,40,41}.

A key benefit from backward contact tracing is the fact that it helps identifying the source that infected the index case in question. This in turn can help isolating micro outbreaks. Apart from looking at the crowded events, the paper did almost nothing to infer chains of transmission. Inferring these chains can help understanding the role of timely testing as well as identifying potential superspreaders.

We indeed did not infer transmission chains on an individual level in this manuscript. All analyses are focused primarily on questions related to decision making during contact tracing: “who should be quarantined and tested and what benefit can be expected?”.

In additional analyses however, we show to which generation backward and forward traced contacts likely belong on a group level (*Figure 5, panel a*). As the main benefit of backward contact tracing is to be derived from the quarantining of sibling cases – who are generally still infectious, as opposed to parent cases - the fact that mostly sibling cases are identified directly in the extended contact tracing window is encouraging.

It is true that micro-outbreaks are the mainstay in our database. In response to the reviewer’s comment, we performed an additional analysis to quantify overdispersion in our dataset. We demonstrate in *Supplementary Figure 4* that the distribution of individual secondary attack rates indeed corresponds to literature. When fitting a negative binomial distribution to individual secondary attack rates, the mean number of secondary cases R_t is shown to be 0.595 and the overdispersion parameter “ k ” is shown to be 0.407.

Backward contact tracing can result in recommending more individuals to quarantine than forward

contact tracing, which may have severe social and economical impact. At the same time, one can argue that backward contact tracing can lead to less quarantining overall. Again, the paper could have tried to explore this a bit more.

Our analyses are indeed focused on empirical data to inform individual level decision making during contact tracing. We have therefore not modelled on a population scale the influence of our approach on the epidemic trajectory, the overall use of testing resources or quarantine burden. We have clarified this focus on the individual level further in the discussion: "Overall, our results show that, on an individual level, the immediate cost and burden of backward contact tracing are proportional to the additional cases identified. Our data do not allow inferences about the impact on a population scale. However, several modelling studies have suggested that the improved epidemic control offered by backward contact tracing has the potential for dramatically lower costs to society, in the form of reduced testing, quarantine and illness^{5,10-12}."

We explored the testing and quarantine burden placed on backward and forward trace contacts in the analysis on contact testing (see above), showing that backward traced contacts can frequently be released from quarantine following a single test immediately after identification.

The study excluded contacts (324 contacts) who were already exposed to another diagnosed index case in the seven days leading to the contact tracing interview. This is understandable from a contact tracing point of view, but I wonder whether these cases were in the end somehow related (i.e. part of the same micro-outbreak)?

We have further clarified this particular matter of dual exposures in the methods section: "Individuals who were already identified as contacts exposed to a previously diagnosed index case within 7 days before the contact tracing interview were excluded as contacts from the second identified index case, while still being considered as contacts for the first. While this approach introduces ambiguity as to the exact day of last exposure, it is reflective of our focus on decision making at the time of first identification of a contact." We would like to refer again to the computed overdispersion parameter, which indeed demonstrates a similar level of clustering in the dataset as observed in other published contact tracing datasets.

I liked the analysis of source events, however, defining whether a person was infected because of a particular source event needs more sharpening. This question is essentially a timeline question. Whether an infection can be causally linked to a crowded event depends on whether the index was infected at the event and on when the index did pass on the infection.

The question of what constitutes a source event and whether a particular contact was present or absent at this event was also approached from the point of view of a contact tracer required to make a decision on who to quarantine and test based on a contact tracing interview. Suspected source events were defined as follows: "Suspected source events were defined as events which, at the time of the contact tracing interview with the index case, were identified as the likely source of the infection, because the index case knew that an individual was present with a confirmed infection or suggestive symptoms. If the index case had been in quarantine since travelling from abroad, travel was considered the source event and travel companions were considered present. Multiple suspected source events were taken into account per index case if applicable. Suspected source events were required to fall within the backward tracing window at least partly to be labelled as such." We have added this last – time related aspect – to the definition in the methods section. The timeline aspects are explored in the previously mentioned additional analyses.

The high positivity rate is quite interesting and I would have liked to see more analysis that can help shed light on the underlying mechanisms. Potential explanations as the authors mentioned is common sources of infection, index cases that were contagious before the symptoms onset, etc. The dataset can probably help isolating some of these causes by, for example, examining differences between individual cases as well as transmission chains.

Our additional analyses seem to imply that a common source of infection (for example a shared source event) is the main mechanism which causes the high positivity rate in backward traced contacts. Two additional graphs support this theory: superspreading individuals, as shown by overdispersion in the distribution of secondary infections (*Supplementary Figure 4*) and a high proportion of sibling rather than parent cases amongst backward traced contacts (*Figure 5, panel a*). We have elaborated on this in the discussion section.

Reviewer #3 (Remarks to the Author):

Raymenants et al collected and analysed SARS-CoV-2 contact tracing data with a view to evaluating the potential for improvements through 'backward' contact tracing: identifying the person who infected an index case, rather than the more usual identification of people the index case infected ('forward' tracing). The data gathered is appropriate, albeit in a subpopulation of higher-education students with limited generalisability, as the authors already concede. The analyses are simple but appropriate, essentially calculating fractions within different subsets of the data. The writing is clear and accessible.

Journal guidance to reviewers asks "do you feel that the results presented are of immediate interest to many people in your own discipline and/or to people from several disciplines?" It also states that "a paper should represent an advance in understanding likely to influence thinking in the field, with strong evidence for their conclusions. There should be a discernible reason why the work deserves the visibility of publication in a Nature Portfolio journal rather than the best of the specialist journals." I think that these results will only be of interest to people focussed on COVID-19 contact tracing, and novelty has been overstated, and I haven't identified a discernible reason for publication outside of the best of the specialist journals. On the other hand, the results are useful, and the topic of COVID-19 contact tracing is currently important for public health, and I agree with the authors that starting the tracing window at least a little bit earlier and investigating source events are both warranted. I therefore defer more than usual to the editor's discretion on acceptance.

The introduction focuses on backward contact tracing as a concept distinct from forward tracing, as supported by previous theoretical work, and states that no study until now has evaluated this approach. I think this framing overstates the novelty of the current study. An alternative lens through which these results can be considered is as follows. The authors identified contacts linked to index cases, calculated the positivity rate among these individuals (the fraction of them testing positive, usually called the secondary attack rate in such studies) when disaggregating by different characteristics or covariates, in order to see which of these are associated with greater risk of transmission. This is a study design that has been widely used and reported throughout the pandemic, for example Koh et al published a meta-analysis of 57 such studies in 2020

<https://journals.plos.org/plosone/article?id=10.1371/journal.pone.0240205>. As the authors note, backward contact tracing in

practise is either through (A) extending the tracing window further back in time than the usual 2 days before the case's symptoms or diagnosis, and/or (B) explicitly attempting to determine the event at which the case became infected - "source investigation". Policy decisions on the two actionable points A and B are informed by the two main quantities measured here: (a) how likely contacts are to test positive as a function of when they were exposed relative to the case's symptoms or diagnosis, and (b) how likely contacts are to test positive depending on whether they were at the suspected source event. Previous studies measuring (a), even if they did not discuss the possibility of a sibling rather than a parent-child relationship for transmission, include

<https://www.nature.com/articles/s41591-020-0869-5>

<https://jamanetwork.com/journals/jamainternalmedicine/fullarticle/2765641>

<https://www.medrxiv.org/content/10.1101/2020.09.04.20188516v2>

We agree that our analysis focusing on positivity rates is relatively simple, and similar to previous assessments of secondary attack rates based on the time of exposure. However, we would like to emphasize that none of these references or other studies published to our knowledge determined the exact exposure window for a sufficiently large group of case-contact pairs to derive the positivity rate (SAR) based on the last day of exposure. The positivity rate (SAR) by last day of exposure is the parameter we view as essential to determine how far back to extend the contact elicitation window. To our knowledge, our study, designed from the outset specifically to assess the additive benefit of

extended window contact tracing, is still the first to supply this piece of information, which is crucial in determining the efficiency of backward contact tracing.

The authors note that source investigation is already used by several countries, citing four references in support of it, to which could be added its recommendation by WHO https://apps.who.int/iris/bitstream/handle/10665/339128/WHO-2019-nCoV-Contact_Tracing-2021.1-eng.pdf?sequence=24&isAllowed=y and the US CDC <https://www.cdc.gov/coronavirus/2019-ncov/php/contact-tracing/contact-tracing-plan/source-investigation.html>. The results here could therefore be alternatively framed as further measurement of the relationship between risk and time of exposure, and further evidence in support of source investigation, which would be less striking but more fair than the first-study-of-its-kind claim made currently.

We maintain that this study is the first to empirically study the efficiency of backward contact tracing. However, we agree that our initial conclusion was worded too strongly. We have changed the conclusion in the discussion section and abstract as detailed below.

In response to this and other reviewers' comments, we also dug deeper into details of backward traced contacts. We added data showing the relative position of contacts in the transmission tree, the level of overdispersion in our dataset and the expected effects on testing and quarantine costs. Lastly, we repeated our main analysis in different time periods, which differed in terms of the dominant circulating VOC, population immunity, the stringency of general contact restrictions, the level of community transmission of COVID-19 and the government requirements concerning testing and quarantine. While conclusions in these follow up periods should be treated with caution due to lower follow-up rates, we decided to add them to the current manuscript as they may provide complementary insights into the underlying dynamics and effects of backward contact tracing.

As suggested, we added references to the recommendations by WHO and CDC.

Major suggestions

The paper's conclusion as phrased in the abstract - "Our results imply an urgent need to implement backward contact tracing globally" - is by some way too strong based on the evidence presented here. One of five limitations acknowledged by the authors is

265 "index cases were young adults in tertiary education, whose socio-economic status
and contact patterns may differ significantly from other age and social groups, limiting
generalisability"

This study provides sufficient motivation not for immediate implementation of policy globally, but for further pilot studies to be conducted by contact tracing teams operating at different hierarchies - local, regional, national - with more representative populations and in different geographic, socio-economic and epidemic contexts. By testing the extra contacts identified by this procedure, without imposing a requirement to quarantine, the main cost of the intervention (additional unnecessary quarantine) can be avoided while gathering more evidence for evaluation. One would therefore hope to find a good appetite for embedding such pilot studies in existing tracing programmes.

We acknowledge that our conclusion was worded too strongly. We have changed the wording to provide a more nuanced view, taking the above suggestions into account.

In the abstract, we state "Our results support implementing backward contact tracing when rigorous suppression of viral transmission is warranted."

In the discussion section, we state: "Our results indicate that in the context of significant community transmission of COVID-19 and in the presence of moderate contact restrictions, there can be a marked added benefit, at low relative cost, to extending the contact tracing window backward beyond the infectious period of the index case."

The paper's advocacy of backward tracing is based only on the number or fraction of contacts identified who are infected. The extent to which contact tracing reduces transmission from each of these individuals depends on two factors: how much they reduce their infectiousness after they are traced (e.g. by isolation) and when they were traced relative to their infectious period (i.e. tracing them a fraction x of the way through their infectious period means a fraction $1-x$ is left to be affected by the intervention). The first point is sufficiently obvious as to not need special mention here. The second point is critical even for purely forward tracing, due to the short generation time and pre-symptomatic infectiousness of SARS-CoV-2 infections. It is more important still for backward tracing: going backwards in the chain of transmission means infected contacts identified this way will be identified later in their infectious period. If they are typically identified toward the end of their infectious period, tracing them will have little effect. Bidirectional iterative tracing will mitigate this to some extent, but I expect the average effect will always be a later identification of contacts than in purely forward tracing. The authors' data could quantify this effect in two ways. Firstly, by comparing the time at which contacts were traced to the time of their onset of symptoms, as a function of when the contact was exposed relative to the index case's symptoms. The general trend should be that as we move earlier into the extended tracing window towards -7 days, the later the tracing will be relative to the contact's symptoms (and by implication, relative to their infectious period). Secondly, by comparing the time at which contacts were traced to the time of the suspected source event, for those contacts who also attended. If neither of these time intervals can be obtained from the data, the general point concerning timings should be acknowledged as a caveat for this evaluation and for backward tracing generally.

We acknowledge that, as mentioned by multiple reviewers, infected contacts in the backward contact tracing window risk being further into their infectious period when detected, which would reduce the effect of their testing and quarantine on onward transmission.

Therefore, we added the analysis the reviewer proposed, comparing the delay period between the detection of an index case (as a proxy of their tracing date) on the one hand and the onset of symptoms of their infected symptomatic contacts on the other. The result is that backward traced contacts were indeed detected on average 1.8 days later in their infectious cycle than their forward traced counterparts, which could be interpreted as a reduction in contact tracing efficiency equal to an additional testing or tracing delay of the same period. By comparing the onset dates of cases and their contacts, we further show that the backward traced group consisted mostly of sibling cases rather than parent cases. The results are discussed under the new results section "Backward contact tracing identifies mostly sibling cases" and visualized in *Figure 5* in the main text. However, as backward traced contacts were last exposed to the index case on average 4 days earlier than their forward traced counterparts, their testing and quarantine burden was smaller, which could again improve the cost-benefit ratio of the approach (see new results section "Less tests and shorter quarantine in the backward traced group", *Figure 6* and *Supplementary Figure 5*).

"The data that support the findings of this study are available on request from the
corresponding author (Joren Raymenants). The data are not publicly available because they
could compromise research participant privacy."

The raw data at point of collection is of course sensitive and should not be public. However, after it has been processed and simplified to the point of entry in the main analysis, it consists of only the following variables for each traced contact:

- was their outcome: a positive test, a negative test, or lost to follow up;
- how many days were in between their last exposure to the index case and the index case's onset of symptoms or diagnosis (note: neither of the two dates is required, only their difference);
- was the contact at the case's suspected source event: yes, no, or source event not identified;
- what kind of relationship is there between case and contact: friend, family etc.

These data do not permit identification of the individuals in question, and would allow reproduction of the authors' main results from the data (and possible extensions, for example fitting a smooth

parametric function to the transmission risk over subsequent days). Inclusion of the data in this form as supplementary material is recommended. Additionally, it was reported that “72.5% of index cases self-reported being symptomatic at the time of testing”. Simply adding a binary variable for each contact noting whether they were associated with these cases would provide partial discrimination between the cases’ symptomatic infectious phases and their pre-symptomatic/asymptomatic phases. Additionally noting when the index did develop symptoms relative to their diagnoses (again, without actual dates, just differences) would allow full discrimination of these phases. Quantifying the difference in transmission risk between these phases is an important epidemiological issue extending beyond contact tracing.

For data that is not released with the paper, the criteria that those wishing to access it must satisfy should be stated here. For example a demonstrable affiliation with an academic or health institution, a legitimate epidemiological question, commitment to not attempt to de-anonymise etc.

In response to this comment, we have anonymised the data and submitted it with the updated manuscript. We have also altered the data availability statement as follows: “The data underlying the main analysis in this manuscript are available in the article and in its online supplementary material. The data that is not released with the paper can be made available on request from the corresponding author (Joren Raymenants) in case of a demonstrable affiliation with an academic or health institution, a legitimate epidemiological question and a commitment to not attempt to de-anonymise.”

Contact tracing is a cost-benefit trade-off. The authors acknowledge this to some extent, implicitly, by basing their recommended expansion of the tracing window upon the observation that it has minimal effect on the efficiency of identifying infected individuals (i.e. similar positivity rates in the two traced groups). Some explicit acknowledgment of the trade-off would be welcome, both in the introduction and the discussion. For example in the introduction, one paragraph discusses what previous modelling studies say about backward tracing, but only with regards to its benefits. Did these studies comment on the costs? For example the fraction of backward traced individuals who are not actually infected, or how much extra quarantine is required for the epidemiological gains? Did these studies address the previous paragraph's point concerning speed? A cost to be acknowledged in the discussion is that, even if backward tracing identified infected individuals with exactly the same efficiency as forward tracing, tracing more individuals requires more work if done manually. Expanding tracing therefore requires hiring additional contact tracers, and/or reduces the maximum number of index cases manual tracing can handle.

We acknowledge the first manuscript lacked details on the cost-benefit trade-off of the proposed contact tracing approaches. The aspects of timely contact detection, quarantine burden and testing requirements are now elaborated on more extensively throughout the discussion. E.g.: “The main trade-off was that infected backward traced contacts were identified on average 1.8 days later in their infectious cycle than forward traced contacts. The burden of testing and quarantine was lower in backward traced contacts due to inherent differences in the timing of their last exposure to the index case.”

Lastly, we acknowledge in a new paragraph in the discussion that our results do not allow direct inferences about the cost-benefit ratio of the expanded testing and contact tracing approach on a population level: “Overall, our results show that, on an individual level, the immediate cost and burden of backward contact tracing are proportional to the additional cases identified. Our data do not allow inferences about the impact on a population scale. However, several modelling studies have suggested that the improved epidemic control offered by backward contact tracing has the potential for dramatically lower costs to society, in the form of reduced testing, quarantine and illness^{5,10-12}.”

With regards to the specific question of whether the referenced studies addressed the previous paragraph's point concerning speed:

- Endo et al did not formally take delay times into account. They assumed the parent case to be no longer infectious at the time of tracing, making the effectiveness of backward tracing solely reliant on the ability to subsequently quarantine infected sibling cases.

- Bradshaw et al did not perform separate analyses in which testing and tracing delays were altered.
- Kojaku et al did not explicitly account for variations in testing and tracing delays. They integrated both into a failure probability of contact tracing per node, for which published parameters are used as inputs.
- Fyles et al chose standard testing and tracing delays in their backward contact tracing scenario but did note that testing delays in general had a particularly strong influence on the impact of contact tracing.

An important distinction between these studies and the results we describe is that all modelling studies conceive backward contact tracing as a two-step process in which the only way to identify sibling cases is through identification of the parent case and their secondary contacts. Our results highlight however that many sibling cases can be identified directly as contacts of the index case, which reduces the effect of delay times in testing and tracing. This has been clarified in the discussion: “Previous studies have emphasized that the benefits of backward contact tracing hinge on the ability to identify first the parent case and then sibling cases in a two-step process, which is likely to be highly susceptible to testing and contact tracing delays^{5,10-12}. However, the distribution of differences in symptom onset dates between index cases and their backward traced contacts suggests that most backward traced contacts were sibling cases (*Figure 5, panel a*) identified as direct contacts of the index case, without the need to first identify the parent case.”

Minor suggestions

The authors could comment on possible differences in implementation between digital and manual contact tracing. For example with digital tracing, expanding the window of exposure would only require changing the value of a parameter of the algorithm, not an expansion in the tracing team's workload; however, interviewing the case about their suspected source event is not possible. The NHS COVID-19 App in England and Wales sends 'risky venue' warning notifications to those who anonymously 'checked in' with the app to a venue that later becomes linked to enough cases. In our view, the changes required to implement an extended contact tracing window are equally minor for digital and manual contact tracing. The additional workload for manual contact tracers should logically be no more than proportional to the additional number of (similarly high-risk) contacts identified. We have clarified this in the following (altered) statement in discussion: “Our approach was to extend the contact tracing window back in time from 2 to 7 days before symptom onset or test, and to systematically refer all identified close contacts in this period for testing, as well as co-attendees of high-risk events. This simple change in standard protocol, which could be implemented both in manual and digital contact tracing, allowed mostly sibling cases to be identified quickly as direct contacts of the index case.”

The point of reference for comparison of positivity rates is individuals who have self-referred for testing based on symptoms. It should be acknowledged that the positivity rate in such a control group could vary widely depending on the incidence of SARS-CoV-2 infections in that place, time and subpopulation, and the incidence of other respiratory illnesses at that place, time and subpopulation. We further clarify this fact in statements under discussion: “Positivity rates amongst symptomatic individuals are dependent on many factors, such as the level of community transmission of SARS-CoV-2 and other respiratory viruses. Still, this group was chosen as a control group, because it represents a high bar and testing of symptomatic patients is standard in most protocols globally.” and “The control group probably also suffered a further reduction in reliability after the main study period, due to the rollout of alternative testing methods such as pharmacy-based and self-administered rapid antigen tests and the progressive scaling back of RT-qPCR testing in general^{37,45,46}.”

The authors refer to "an incubation period" / "the incubation period" without acknowledging variability in this quantity between individuals, as captured by the incubation period distribution. Since

backward tracing is based on going back in time by one incubation period, at least a short discussion of this distribution is warranted. The greater the variance of the distribution, the harder it is to estimate when an individual was infected.

The variability in the incubation phase is now noted when the rationale for backward contact tracing is first introduced under introduction: "Another approach is to extend the contact tracing window back in time and to systematically refer all close contacts for quarantine and/or testing (*Figure 1, 2*). This assumes that, if the tracing window is extended backward by at least the incubation period of the index case, the parent case can be identified, as well as sibling cases present at a shared source event. To this end, the contact tracing window should be extended far enough to include most of the variability in incubation periods³⁴"

The authors already note that the impact of backward contact tracing is linked to overdispersion in the offspring distribution (the number of individuals to which each infected individual transmits). Since transmission risk can be decomposed into the product of biological infectiousness (the rate of shedding viable virions) and the rate & intensity of contact events with susceptible individuals, the overdispersion in actual transmission events can be decomposed into contributions from overdispersion in biological infectiousness (with cross-sectional viral load surveys being one proxy) and overdispersion in social contact networks. Where either of these factors is expected to be variable depending on the subpopulation - and presumably the second factor at least is - this provides a specific example of the stated limitation about the representativeness of higher education students as a study group. Which could be commented on, if desired.

As mentioned by the reviewer, the particularities of the contact patterns of the target population and setting are commented on under the discussion section: "First, it took place in the setting of moderate general contact restrictions, which altered social patterns significantly and likely increased the efficiency of identifying source individuals by decreasing the number of contacts in general and casual contacts in particular, which are harder to identify through manual contact tracing. Second, index cases were young adults in tertiary education, whose socio-economic status and contact patterns may differ significantly from other age and social groups, limiting generalisability⁴⁸"

When discussing the possible explanations for a high positivity rate observed in contacts last seen before the contagious period, the concept of overdispersion is now discussed further: "This explanation is supported by the fact that secondary infections linked to a particular index case followed a binomial distribution with a value for the dispersion parameter k of 0.407 (*Supplementary figure 4*), which implies a degree of overdispersion corresponding with literature^{20,22,23}. The finding that backward traced contacts seem to consist mostly of sibling rather than parent cases, also supports this mechanism (*Figure 5*)."

Supplementary Figure 4 plots the distribution of infected contacts per index case, from which R_t and k are derived by fitting a negative binomial distribution.

"Case-based interventions such as case isolation, contact tracing and quarantine"

Quarantine is not always a case-based intervention: it is for those at risk of being infectious for some reason, for example travel. I suggest "such as case isolation or contact tracing with quarantine" instead.

The sentence was adjusted as suggested.

"In addition to child cases, any practical
forward tracing strategy probably identifies the parent case (the infector of the index case)
and sibling cases (infected by the same parent case) some of the time"

It would be good to clarify here why this happens, because the frequency with which it happens determines the additional benefit of backward-by-design tracing. i.e. either the case has a second contact with their infector during their own infectious period, or the time from the case's infection to their symptoms or diagnosis was less than two days.

We have altered the sentence accordingly: “In addition to child cases, any practical forward tracing strategy probably identifies the parent case (the infector of the index case) and sibling cases (infected by the same parent case) some of the time, for example if the index case had repeated contact with their parent or sibling case during their own infectious period, or if the time from the index case’s infection to their symptom onset or diagnosis was less than two days¹².”

Similarly, under the backward contact tracing section of the introduction, we have added: “In any practical implementation, additional child cases may also be identified through backward contact tracing, for example if the index case’s infectiousness started more than two days before symptom onset¹²”

The fact that we did not make assumptions on the directionality of transmission is stressed in the introduction and results sections. We explore the generations to which forward and backward contacts mainly belong under “Backward contact tracing identifies mostly sibling cases” and in the new *Figure 5, panel a*.

The authors mention that tracing iteratively can improve reach, in the sentence after backward tracing is first introduced. I suggest acknowledging very briefly that iterative tracing extends the reach of purely forward tracing too, lest it be mistaken as something specific to backward or bidirectional tracing.

This is indeed the case. We have added the possibility of iterative tracing for both the backward and the forward tracing group in a new overview figure (*Figure 1*). We also stress in the introduction that “Due to these delays, immediate testing of identified contacts in support of iterative tracing may be especially relevant in backward contact tracing.” The importance of the timing of contact testing and its resulting influence on contact tracing efficiency is further explored under the “Timing of contact testing” section under results, in the discussion section and in *Figure 6* and *Supplementary figure 5*.

"High
positivity rates are reported for attendants of such source events"

The phrasing sounds like positivity is high at all events with suspected transmission, but sometimes the suspicion will be wrong. I suggest "have been reported... at some source events"

The sentence is now “High positivity rates have been reported for attendants of some source events³³”

"This approach [source investigation] is reliant on
the identification of multiple infected cases at the same event, by pooling of contact tracing
data from different index cases"

I assume the authors mean in practice but not in theory, because even just a single case can speculate on which event they were infected at, and if I have understood the results presented here that is precisely what happened. If the statement above applies to practice, its scope should be clarified and supporting evidence cited. I suspect that there will be many contexts where this is not true - where contact tracing teams investigate events where a single case might have been infected, without (or before) a second case being linked to that event.

The sentence was nuanced and changed to the following: “In practice, this approach is usually reliant on the identification of multiple confirmed or probable infected cases at the same event, for example by pooling of contact tracing data from different index cases or asking the index case about other cases in their environment. As a result, the approach can fail to identify the source event at the time of identification of the initial index case.”

The analysis on source events in this manuscript was conceived as the possibility – in practice – of an experienced contact tracer to identify a putative source event at the time of the initial contact tracing interview with an index case. The definition has been further clarified as follows: “Suspected source events were required to fall within the backward tracing window at least partly to be labelled as such.”

"both as soon as possible and after an
incubation period."

Confusing phrasing - the two aspects seem to contradict each other.

We have changed the phrasing as follows: "Another approach is to extend the contact tracing window back in time and to systematically refer all close contacts for quarantine and/or testing (*Figure 1, 2*)". The diagnostic value of contact testing by day post-exposure is further investigated in a separate section "Less tests and shorter quarantine in the backward traced group" under results.

"Fyles et al also show in a branching process model that an extended contact tracing window
results in a linear decrease in the growth rate up until around 8 to 10 days prior to symptom
onset or diagnosis[11]."

An important caveat from reference [11] should be acknowledged:

"However, identified cases might struggle to remember contacts they have had back in the past. We find that including a daily 10% reduction in the probability of recalling a contact in the model erodes the gains of backwards contact tracing almost completely"

We added a sentence stating ", although additional gains are highly reliant on recall decay"

"identified in an extended contact tracing window, starting 7 days before onset of symptoms or
diagnosis."

I suggest adding here "(whichever was earlier)"

The phrase was added as suggested.

"at the time of writing"

I suggest replacing by a specific moment in time for ease of reference in future.

The sentence was changed to "Our test and trace program started in September 2020 and is still active in April 2022".

"We did not collect demographic data on contacts of index cases..."
There was a slightly higher percentage of women in the control group (56.5%, missing data
3.0%) while the mean age was similar to the other groups"

To which groups is the control group being compared to here? I interpreted line 163 as meaning demographic data is unavailable for the standard tracing window group and the study group.

The demographic data of cases are being compared here. We have changed the sentence as follows: "There was a slightly higher percentage of women in the control group (56.5%, missing data 3.0%) compared to the index cases, while the mean age was similar (22.0 years; SD 3.84 years, missing data 3.0%). The temporal distribution of individuals in the backward traced contacts, forward traced contacts and symptomatic control groups is shown in *Supplementary Figure 1*."

"backward contact tracing..."
identifies an additional 43.5%
of cases"

Use of the present tense suggests this is a universal truth. Past tense, and restriction of the sentence to the present study, would clarify the scope.

We have changed the phrase to the past tense: "This study lays out a strategy for backward contact tracing which markedly improves the effectiveness of contact tracing in the setting of COVID-19. It identified an additional 42% of cases not detected through the contact tracing protocol used in most jurisdictions, gains which are likely to have a major impact on epidemic control¹²". Since the sentence before starts with this study, we believe it is now clear the number relates to the present study. (Note that the percentage of additional identified cases, which was wrongly presented in this sentence but correctly elsewhere in the manuscript, has been corrected to 42%.)

414 "We excluded contacts already identified as exposed within 7 days before the contact tracing
415 interview to a previously diagnosed index case."

I think this means that such contacts were excluded from the study entirely; however, it would be good to clarify whether they were only excluded as contacts from the second identified index case, while still being considered as contacts for the first. In that scenario, the contact's exposure time is ambiguous without further details.

Indeed, for contacts exposed to multiple index cases the time of last exposure can be ambiguous. However, in this study we were interested in determining the risk of infection for contacts, using only information available at the time of the initial interview with the index case. We decided not to exclude contacts of multiple index cases entirely, fearing this would exclude contacts involved in superspreading events, which constitute an important mechanism in the spread of COVID-19. We've thus altered the sentence as follows: "Individuals who were already identified as being contacts exposed to a previously diagnosed index case within 7 days before the contact tracing interview were excluded as contacts from the second identified index case, while still being considered as contacts for the first. While this approach introduces ambiguity as to the exact day of last exposure, it is reflective of our focus on decision making at the time of first identification of a contact."

A comma is used as a decimal separator in the results figure, whereas a point is used in the prose.
The figure was adjusted to use a point as a decimal separator.

Extended data Figure 1 needs an x axis label.
The figure was adjusted as suggested.

No comments have been provided to the editor separately from those here.

With best wishes to the authors,

Other changes to the manuscript:

- Seven case-contact pairs were excluded from the analysis, who were previously included. The reason is that our procedure led to erroneous inclusions, if a contact was already in quarantine after exposure to a different case at the time of the contact tracing interview and had already been reported as a contact of a third case in the past. This error had no effect on the conclusions of the study.

Best regards to the reviewers,

Joren Raymenants and colleagues

REVIEWER COMMENTS

Reviewer #1 (Remarks to the Author):

I thank the authors for engaging fully with the reviewers. I think they have responded acceptably to all my comments.

Reviewer #2 (Remarks to the Author):

I think the revised version has made reasonable attempts at addressing the reviewers' comments. I like in particular the new analysis of the type of contacts identified by backward contact tracing as well as the cost benefit analysis considering the time of identification.

Although the new additions have improved the manuscript, I still think that the cost-benefit analysis could have been more elaborate. For instance, the model by Endo et al suggests that backward contact tracing can help avert 2-3 times the number of cases that forward contact tracing averts. Now, this paper can try to look at this particular metric by for example quantifying asymptomatic cases that were identified by backward tracing and all subsequent asymptomatic children. To sum up, I would like to see some comparison with modeling studies.

The authors rightly argue that their focus is on informing individual level decision making during contact tracing. However, observations from contact tracing on spreading patterns were used by health authorities as an important input when deciding on pandemic control measures. Hence, going from the individual level to labeling whether an infection is related to a source event, regular daily contacts or a community transmission is important. It would be nice to sketch some ideas on how data from backward contact tracing can be used to construct such feedback.

Minor issues:

The second last sentence in the abstract is hard to read

Figure 6b only has a right y-axis. I am not sure if this is intentional. The common practice is to have a left y-axis when there is only one y-axis

Reviewer #3 (Remarks to the Author):

The authors have largely dealt with the issues raised in the first round of review; in particular, providing the data underlying the analysis is a significant improvement. I believe a little more work is needed to address the main issue raised by all three reviewers before publication. Namely, the expected delay in reaching contacts traced through the extended window (with exposures to the index that were further back into the past) compared to those normally traced. Specifically, the issue is how much this delay would reduce the epidemiological impact of quarantining these individuals, because a smaller fraction of their infectious period remains to be affected by quarantine by the time they are traced. In revision, the authors have quantified this delay - 1.8 days - but have not quantified its major effect, a negative effect on epidemiological impact. Instead they have quantified, and seem to give more emphasis to, the minor positive byproduct of this delay which is reduced quarantine and testing for such contacts. This is a strange choice of emphasis: if we imagine artificially making this delay longer, the minor positive byproduct would become even bigger, but of course eventually the epidemiological benefit of tracing these contacts would disappear, making the intervention redundant. This is the main effect of the delay to focus on.

This effect can be quantified straightforwardly by comparing (a) the observed times at which infected contacts were reached by the tracing team relative to their own symptoms, with (b) previous estimates of how the risk of transmitting to others varies with time with respect to ones own symptoms. Thanks to the authors now providing the data underlying the analysis, I was able to calculate this using one previous estimate of the timing of transmission. The result is a 28% decrease in how much infectiousness remains when the index was diagnosed, comparing the mean

for the extended tracing window to the mean for the standard one. I assumed the delay between diagnosis of a given index case and tracing their contacts was always zero; if the time of reaching contacts was recorded, this would be better to use. I could not see how to attach a file to my review so my code can be accessed here

[REDACTED]

Should the authors wish to conduct their own calculation of this effect instead of the one I have provided, two points require care. First, a given delay, 1.8 days say, has a different effect depending on when in the infectious period it occurs (e.g. if the tracing before introducing the delay was already extremely slow, the delay can't make it much worse). Second, the quantitative relationship between infectiousness and timing is very much non-linear, and so one should calculate the remaining infectiousness at time of being traced for each contact first, and then take any desired averages e.g. for the extended vs standard windows. Reversing the order of these steps - calculating the average timing in each group and then calculating the infectiousness remaining for that average timing - would give the wrong answer. In other words, we want $\langle \text{infectiousness}(\text{timing}) \rangle$ not $\text{infectiousness}(\langle \text{timing} \rangle)$, with $\langle \dots \rangle$ denoting the expectation operation.

MINOR ISSUES

I didn't see a description of the analysis underlying Supplementary Figure 4 in the paper.

"For the main analysis, we did not make assumptions on the directionality of transmission.
Therefore, both the forward and backward traced group likely included parent, sibling and child
cases."

These assumptions would have been required for Supplementary Figure 4, so which transmission pairs were included in this analysis / what were the inclusion criteria? Was transmission direction determined simply as from the individual diagnosed earlier to the one diagnosed later?

Figure 1: the legend says that the transmission from the parent to the index and sibling occurred at a source event, and that attendees of the event are circled, so it seems the parent should be circled as well as the index and sibling.

134: "diagnosis, although additional gains are highly reliant on recall decay"

Suggest rewording to "highly sensitive to recall decay", as reliance means a positive dependence but this one is negative.

Figure 5: "Based on published serial intervals for the Alpha strain of 4-5 days, the arrows roughly indicate the ranges where we expect to find most parent, sibling and child cases."

The relative positioning of the arrows and the data is the basis for the authors' conclusion that backward-traced contacts are mostly siblings (though see the caveat to this inference in the following paragraph). It's therefore desirable to state exactly, not roughly, what the arrows indicate. They correspond to ranges of a probability distribution, but which range, e.g. the 2.5-97.5th percentile range or the 25-75th? I believe a different range is shown for siblings than for the parents or children, which is slightly misleading. (The distribution for siblings should have larger variance, being given by two draws from the serial interval distribution instead of one.) Similarly, please state exactly what serial interval distribution was used for this calculation - including its functional form and variance - not just that the distribution mean was 4-5 days.

The authors observe that the timings for contacts in the extended tracing window fall in the range expected for siblings, not the ranges expected for parents or children, and from this infer that the contacts are mostly siblings. Formally, this is an example of the prosecutor's fallacy: equating $P(\text{timings} | \text{sibling})$ with $P(\text{sibling} | \text{timings})$. The difference between these probabilities depends on the a priori probabilities of contacts being the siblings, children or parent of the index, and these probabilities depend on both R and the overdispersion of the offspring distribution. For example, assume overdispersion is extremely large and R is roughly 1: say 1% of infected people transmit to 100 people each, the rest infecting no-one. This implies every infected individual has exactly 99 siblings, has exactly 1 parent as always, and is expected to have 1 child. Here a given timing could be much more likely when conditioning on being a parent (or a child) than when

conditioning on being a sibling, but this can be outweighed by the number of siblings. At the other extreme, with no dispersion in offspring distribution and $R=1$, each infected individual transmits exactly once and so no-one has any siblings. A full calculation here would probably be overkill, but I encourage the authors to at least lay out their inference and assumptions clearly, to avoid perpetuation of this common fallacy which alas does land people in jail. Note that the reasoning here is independent of philosophical stances on Bayesianism - the 'a priori' probability of being a sibling/child/parent is well defined as a fraction of infected individuals as in the two extreme examples above, not merely as a subjective belief.

268 "Forward traced symptomatic contacts were detected
269 on average 1.77 days earlier in their infectious cycle than their backward traced counterparts,
270 assuming equal testing and tracing delays."

This sentence seems to read roughly as "We measured a tracing delay to differ between the two groups, assuming there was no difference in the tracing delay," i.e. it seems to contradict itself. The 1.77-day delay is the difference between the groups in the time of contact's symptoms relative to the index's diagnosis. The difference in delay assumed to be zero at the end of the sentence I think refers to the delay between diagnosis of the index and tracing the contact. I suggest clarifying this.

274 "Immediate testing after contact identification can accelerate iterative tracing ("test
275 to trace") and – if an asymptomatic contact tests positive – reduce to total duration spent in
276 quarantine and isolation"

Is the statement after the parenthesis based on the assumption (true for this study's time and place?) that the duration of isolation/quarantine after a positive test is shorter than after being contact traced, assuming no symptoms? It would be good to clarify this, as it is counter to general expectation that individuals only suspected of being positive should have a lower isolation burden than those actually known to be positive.

Figure 6a might be easier to understand or more informative if "contacts attending for two tests" were split into two categories - first test and second test - because these have different timings and serve different purposes.

Figure 6b: what exactly are the points and the error bars? Are they the mean, and the mean +/- the sample standard deviation?

Lines 309-311: the properties of VOCs are described collectively and in the present tense as though all VOCs did, and will, exhibit the same changes compared to their preceding variant. The different VOCs we have seen so far have differed from each other and may differ in the future - I suggest making the language more precise.

"Our data show that only 49% more contacts at risk are identified by extending the contact
tracing window backward by 5 additional days. This could be explained by recall decay or by
recurring contacts with the same individuals."

The phrasing makes it unclear why an explanation is needed. If the standard tracing window typically went out to 8 days post-symptoms, getting 49% more contacts from an additional 5 days (+50%) would be as expected. It doesn't, so there is a lower per-day number of contacts added from the extended window: that's the better number to quote before explaining why this might be lower.

380 "the number of child cases observed in"

Suggest instead "the inferred number of child cases in", as this is inference rather than observation.

"Third, due to a proven individual
propensity to shed live virus and an above average number of social interactions, the source
case is likely to have initiated other infections among the index's contacts"

Conditioning on an individual transmitting at least once does not generally imply that they are "likely" (presumably meaning having probability of at least 0.5) to have transmitted at least twice, nor that those other transmissions were "likely" to people who are also contacts of the index, which depends on the structure of the contact network. One way of reading this sentence is that it

might not be true at all, because it is third in a list of mechanisms that might be true or might not be. Another way of reading it is that it is definitely true, but the extent to which it explains the high positivity rate in the extended window is unknown. To guard against confusion here I suggest simply replacing "is likely to" with "could have".

384 "This explanation

is supported by the fact that secondary infections linked to a particular index case followed a
binomial distribution with a value for the dispersion parameter k of 0.407"

It is hard for readers to intuit what a given value of k implies for the quantity that directly supports or refutes this mechanism, namely $P(\text{number transmissions} \geq 2 \mid \text{number transmissions} \geq 1) * P(\text{siblings are contacts of each other})$. The second factor is unknown but we may guess that it is high. The authors have calculated the offspring distribution in order to determine k , so the first factor may be calculated directly.

Also, estimated quantities like k here (and R_t in Supplementary figure 4) should be quoted with their uncertainty. It's best practise to quote figures to a number of significant figures appropriate to the size of the uncertainty. e.g. if the uncertainty in k spans 0.3 to 0.5, the second and third decimal places of k are redundant (and quoting them suggests more precision than is the case).

"backward traced contacts seem to consist mostly of sibling rather than parent cases,
also supports this mechanism (Figure 5). Fourth, more distant relatives in the transmission
tree

could be detected due to wider circulation in an index case's social circle."

The inference that these contacts are mostly siblings was implicitly based on the assumption that they are either siblings, children, or parents: nothing else. If the authors consider that these contacts could be elsewhere in the transmission tree, for accuracy they should not be described as "most likely siblings" but "more likely to be siblings than parents or children".

"These results speak in favour of simply referring all close contacts in the extended tracing
window for testing and quarantine, even if they were not present at the suspected source
event.

Additionally, we show that jurisdictions favouring the implementation of a source investigation
strategy would do well to switch to an extended contact tracing window approach when no
clear source event is identified at the time of the contact tracing interview"

I suggest replacing "Additionally, we show that" by "Alternatively", because the two recommendations differ: the first recommends not bothering with source investigation, and the second recommends how to do it if you do it.

447 "Third, the population was almost entirely unvaccinated during the main
448 study period."

It might be worth stating why this is a limitation - is extrapolating from these findings to a vaccinated population just adding some unknown unknowns, or is a bias expected in one direction?

597: comma for decimal separator

"Contacts were assigned to either the standard tracing window group, a reference group
mirroring standard practice, or to the extended tracing window group, based on when their
last

close contact with the index case took place."

I suggest enclosing "a reference group mirroring standard practice" with either dashes or parentheses, because commas make this seem like there are three groups instead of two.

"In the last period, with the Omicron strain dominant, contacts who did
not develop symptoms or undergo testing in the 7 days after exposure were considered "not
infected"."

Was this outcome considered as lost-to-follow-up for previous viral variants? Why was it re-defined to not infected for Omicron?

With best wishes to the authors,
Chris Wymant

REVIEWER COMMENTS

Reviewer #1 (Remarks to the Author):

I thank the authors for engaging fully with the reviewers. I think they have responded acceptably to all my comments.

Reviewer #2 (Remarks to the Author):

I think the revised version has made reasonable attempts at addressing the reviewers' comments. I like in particular the new analysis of the type of contacts identified by backward contact tracing as well as the cost benefit analysis considering the time of identification.

Although the new additions have improved the manuscript, I still think that the cost-benefit analysis could have been more elaborate. For instance, the model by Endo et al suggests that backward contact tracing can help avert 2-3 times the number of cases that forward contact tracing averts. Now, this paper can try to look at this particular metric by for example quantifying asymptomatic cases that were identified by backward tracing and all subsequent asymptomatic children. To sum up, I would like to see some comparison with modeling studies.

The authors rightly argue that their focus is on informing individual level decision making during contact tracing. However, observations from contact tracing on spreading patterns were used by health authorities as an important input when deciding on pandemic control measures. Hence, going from the individual level to labeling whether an infection is related to a source event, regular daily contacts or a community transmission is important. It would be nice to sketch some ideas on how data from backward contact tracing can be used to construct such feedback.

In response to this comment, we built a simple branching process model to quantify the benefits and costs of iterative tracing with either a standard tracing window or an extended tracing window. The results are added to a new section under results "Iterative contact tracing in a branching process model" and to the discussion.

In our iterative tracing model, we found that the number of detected cases increased by 55% when including backward tracing. This is less than two to threefold, but within the very broad range of relative increases predicted in the model by Endo et al. For example, the following is one of the plausible combinations of parameter values for their model, which predicts exactly the same result: $R=1.2$, $k=0.4$, $b=0.5$, $q=0.6$ and $d=0.2$.

Our model results are not directly comparable to other modelling studies (Bradshaw et al, Fyles et al, Kojaku et al), in which outcomes such as R_t or incidence rate are used.

Minor issues:

The second last sentence in the abstract is hard to read

The sentence has been split in two as follows: “Compared to standard practice, backward traced contacts required fewer tests and shorter quarantine. However, they were identified later in their infectious cycle if infected.”

Figure 6b only has a right y-axis. I am not sure if this is intentional. The common practice is to have a left y-axis when there is only one y-axis

We have moved the y-axis to the left of the graph as suggested.

Reviewer #3 (Remarks to the Author):

The authors have largely dealt with the issues raised in the first round of review; in particular, providing the data underlying the analysis is a significant improvement. I believe a little more work is needed to address the main issue raised by all three reviewers before publication. Namely, the expected delay in reaching contacts traced through the extended window (with exposures to the index that were further back into the past) compared to those normally traced. Specifically, the issue is how much this delay would reduce the epidemiological impact of quarantining these individuals, because a smaller fraction of their infectious period remains to be affected by quarantine by the time they are traced. In revision, the authors have quantified this delay - 1.8 days - but have not quantified its major effect, a negative effect on epidemiological impact. Instead they have quantified, and seem to give more emphasis to, the minor positive byproduct of this delay which is reduced quarantine and testing for such contacts. This is a strange choice of emphasis: if we imagine artificially making this delay longer, the minor positive byproduct would become even bigger, but of course eventually the epidemiological benefit of tracing these contacts would disappear, making the intervention redundant. This is the main effect of the delay to focus on.

This effect can be quantified straightforwardly by comparing (a) the observed times at which infected contacts were reached by the tracing team relative to their own symptoms, with (b) previous estimates of how the risk of transmitting to others varies with time with respect to ones own symptoms. Thanks to the authors now providing the data underlying the analysis, I was able to calculate this using one previous estimate of the timing of transmission. The result is a 28% decrease in how much infectiousness remains when the index was diagnosed, comparing the mean for the extended tracing window to the mean for the standard one. I assumed the delay between diagnosis of a given index case and tracing their contacts was always zero; if the time of reaching contacts was recorded, this would be better to use. I could not see how to attach a file to my review so my code can be accessed here

[REDACTED]

Should the authors wish to conduct their own calculation of this effect instead of the one I have provided, two points require care. First, a given delay, 1.8 days say, has a different effect depending on when in the infectious period it occurs (e.g. if the tracing before introducing the delay was already extremely slow, the delay can't make it much worse). Second, the quantitative relationship between infectiousness and timing is very much non-linear, and so one should calculate the remaining infectiousness at time of being traced for each contact first, and then take any desired averages e.g. for the extended vs standard windows. Reversing the order of these steps - calculating the average timing in each group and then calculating the infectiousness remaining for that average timing - would give the wrong answer. In other words, we want not infectiousness(<timing>), with <...> denoting the expectation operation.

We greatly thank the reviewer for sharing these insights and even the code for an additional analysis. The suggested analysis was used to calculate remaining infectiousness in the Results section "Backward contact tracing identifies cases later in their infection". This method was also incorporated in a branching process model to quantify the costs and benefits of iterative contact tracing approaches. The model is described in a new Results section "Iterative contact tracing in a branching process model". The results of the model are shown in a new main text Figure 8 and Supplementary Figure 8. The model methods are described in the supplementary methods section and the source data and code are added in a supplementary data file.

MINOR

ISSUES

I didn't see a description of the analysis underlying Supplementary Figure 4 in the paper.
185 "For the main analysis, we did not make assumptions on the directionality of transmission.

186 Therefore, both the forward and backward traced group likely included parent, sibling
and child
187 cases."

These assumptions would have been required for Supplementary Figure 4, so which transmission pairs were included in this analysis / what were the inclusion criteria? Was transmission direction determined simply as from the individual diagnosed earlier to the one diagnosed later?

No assumptions were made on the directionality of transmission to produce Supplementary Figure 4. The figure simply shows the total number of identified infected contacts per index case. This has been clarified in the Figure legend: "Supplementary Figure 4 shows the distribution of the number of infected contacts per index case, using only the included case-contact pairs in the main study period. This is different from the offspring distribution, as no assumption is made on the direction of transmission."

Figure 1: the legend says that the transmission from the parent to the index and sibling occurred at a source event, and that attendees of the event are circled, so it seems the parent should be circled as well as the index and sibling.

For clarity, the dotted line to indicate the source event has been removed. Instead, we have noted in the legend that the index and sibling cases have a shared source event.

134: "diagnosis, although additional gains are highly reliant on recall decay"
Suggest rewording to "highly sensitive to recall decay", as reliance means a positive dependence but this one is negative.

The wording has been changed as suggested.

Figure 5: "Based on published serial intervals for the Alpha strain of 4-5 days, the arrows roughly indicate the ranges where we expect to find most parent, sibling and child cases." The relative positioning of the arrows and the data is the basis for the authors' conclusion that backward-traced contacts are mostly siblings (though see the caveat to this inference in the following paragraph). It's therefore desirable to state exactly, not roughly, what the arrows indicate. They correspond to ranges of a probability distribution, but which range, e.g. the 2.5-97.5th percentile range or the 25-75th? I believe a different range is shown for siblings than for the parents or children, which is slightly misleading. (The distribution for siblings should have larger variance, being given by two draws from the serial interval distribution instead of one.) Similarly, please state exactly what serial interval distribution was used for this calculation - including its functional form and variance - not just that the distribution mean was 4-5 days.

As suggested, we formally derived the indicated ranges. We used a normal distribution fitted in a published re-sampling meta-analysis (mean 4.87, standard deviation 4.79, <https://doi.org/10.1177/09622802211065159>). We show the 25th-27th percentile ranges, with the range for sibling cases being wider by a factor of $\sqrt{2}$. Unfortunately, we could not identify a publication describing the distribution of serial intervals specifically for the Alpha strain, so we used the above distribution as several articles reported alpha to have a similar a serial interval to the ancestral strain (<https://doi.org/10.12688/wellcomeopenres.16974.2>, <https://doi.org/10.3390/microorganisms9112371>).

The authors observe that the timings for contacts in the extended tracing window fall in the range expected for siblings, not the ranges expected for parents or children, and from this infer that the contacts are mostly siblings. Formally, this is an example of the prosecutor's fallacy: equating $P(\text{timings} \mid \text{sibling})$ with $P(\text{sibling} \mid \text{timings})$. The difference between these probabilities depends on the a priori probabilities of contacts being the siblings, children or parent of the index, and these probabilities depend on both R and the overdispersion of the offspring distribution. For example, assume overdispersion is extremely large and R is roughly 1: say 1% of infected people transmit to 100 people each, the rest infecting no-one. This implies every infected individual has exactly 99 siblings, has exactly 1 parent as always, and is expected to have 1 child. Here a given timing could be much more likely when conditioning on being a parent (or a child) than when conditioning on being a sibling, but this can be outweighed by the number of siblings. At the other extreme,

with no dispersion in offspring distribution and $R=1$, each infected individual transmits exactly once and so no-one has any siblings. A full calculation here would probably be overkill, but I encourage the authors to at least lay out their inference and assumptions clearly, to avoid perpetuation of this common fallacy which alas does land people in jail. Note that the reasoning here is independent of philosophical stances on Bayesianism - the 'a priori' probability of being a sibling/child/parent is well defined as a fraction of infected individuals as in the two extreme examples above, not merely as a subjective belief.

We acknowledge this limitation by changing the wording in the Results section: “Although we cannot ascertain the relative positions of infected contacts in the transmission tree, the observed timings would be consistent with a higher fraction of sibling cases in the backward traced group and a minority of parent cases in both groups.”

We also explain the limitations of this interpretation in the Discussion section: “This inference of relative positions in the transmission tree should be interpreted with caution. The observed differences in onset time between index cases and their contacts are dependent on a priori probabilities of contacts being sibling, parent or child cases. These probabilities are in turn dependent on the reproduction number, overdispersion of the offspring distribution and probabilities of sibling cases being direct contacts of an index case.”

"Forward traced symptomatic contacts were detected
on average 1.77 days earlier in their infectious cycle than their backward traced
counterparts,
assuming equal testing and tracing delays."
This sentence seems to read roughly as "We measured a tracing delay to differ between the two groups, assuming there was no difference in the tracing delay," i.e. it seems to contradict itself. The 1.77-day delay is the difference between the groups in the time of contact's symptoms relative to the index's diagnosis. The difference in delay assumed to be zero at the end of the sentence I think refers to the delay between diagnosis of the index and tracing the contact. I suggest clarifying this.

The wording has been clarified as follows: “..., assuming equal delays between index case diagnosis and tracing of the contact”

"Immediate testing after contact identification can accelerate iterative tracing (“test
to trace”) and – if an asymptomatic contact tests positive – reduce to total duration
spent in 276 quarantine and isolation"
Is the statement after the parenthesis based on the assumption (true for this study's time and place?) that the duration of isolation/quarantine after a positive test is shorter than after being contact traced, assuming no symptoms? It would be good to clarify this, as it is counter to general expectation that individuals only suspected of being positive should have a lower isolation burden than those actually known to be positive.

Actually, this statement was based on the assumption that release from isolation occurs at a set interval after diagnosis, which in our setting was true only for asymptomatic cases. The sentence was clarified as follows: "Immediate testing after contact identification can accelerate iterative tracing ("test to trace"). It can also reduce to total duration spent in quarantine and isolation, in settings where release from isolation is dependent on the timing of diagnosis".

Figure 6a might be easier to understand or more informative if "contacts attending for two tests" were split into two categories - first test and second test - because these have different timings and serve different purposes.

We have change the figure to indicate the number of contacts attending for a first test and the number of contacts attending for a second test. This seems logical because we are dividing the latter by the former to obtain the ratio of contacts returning for a second test.

Figure 6b: what exactly are the points and the error bars? Are they the mean, and the mean +/- the sample standard deviation?

The points indicate the mean, and the error bars indicate 95% confidence intervals. This has been clarified in the legend.

Lines 309-311: the properties of VOCs are described collectively and in the present tense as though all VOCs did, and will, exhibit the same changes compared to their preceding variant. The different VOCs we have seen so far have differed from each other and may differ in the future - I suggest making the language more precise.

The description of VOCs has been changed as follows: "First, increased intrinsic transmissibility may have rapidly overwhelmed the public health system. Second, shortened incubation periods and serial intervals possibly outpaced the delays inherent in testing and tracing."

347 "Our data show that only 49% more contacts at risk are identified by extending the contact

tracing window backward by 5 additional days. This could be explained by recall decay or by
recurring contacts with the same individuals."

The phrasing makes it unclear why an explanation is needed. If the standard tracing window typically went out to 8 days post-symptoms, getting 49% more contacts from an additional 5 days (+50%) would be as expected. It doesn't, so there is a lower per-day number of contacts added from the extended window: that's the better number to quote before explaining why this might be lower.

We changed the phrasing as follows: "Our data show that 49% more contacts were reported when extending the contact tracing window by 5 days. As the contact tracing window was extended backward, fewer additional contacts were identified per day. This could be explained by recall decay, but also by recurring contacts with the same individuals.

Household contacts, for example, were often excluded from the backward tracing group because they were also exposed in the standard contact tracing window."

"the number of child cases observed in"

Suggest instead "the inferred number of child cases in", as this is inference rather than observation.

The phrasing was changed accordingly.

"Third, due to a proven individual
propensity to shed live virus and an above average number of social interactions, the source
case is likely to have initiated other infections among the index's contacts"

Conditioning on an individual transmitting at least once does not generally imply that they are "likely" (presumably meaning having probability of at least 0.5) to have transmitted at least twice, nor that those other transmissions were "likely" to people who are also contacts of the index, which depends on the structure of the contact network. One way of reading this sentence is that it might not be true at all, because it is third in a list of mechanisms that might be true or might not be. Another way of reading it is that it is definitely true, but the extent to which it explains the high positivity rate in the extended window is unknown. To guard against confusion here I suggest simply replacing "is likely to" with "could have".

The phrasing was changed accordingly.

384 "This explanation

is supported by the fact that secondary infections linked to a particular index case followed a
binomial distribution with a value for the dispersion parameter k of 0.407"

It is hard for readers to intuit what a given value of k implies for the quantity that directly supports or refutes this mechanism, namely $P(\text{number transmissions} \geq 2 \mid \text{number transmissions} \geq 1) * P(\text{siblings are contacts of each other})$. The second factor is unknown but we may guess that it is high. The authors have calculated the offspring distribution in order to determine k , so the first factor may be calculated directly.

Also, estimated quantities like k here (and R_t in Supplementary figure 4) should be quoted with their uncertainty. It's best practise to quote figures to a number of significant figures appropriate to the size of the uncertainty. e.g. if the uncertainty in k spans 0.3 to 0.5, the second and third decimal places of k are redundant (and quoting them suggests more precision than is the case).

In Supplementary Figure 4, we intend to show the distribution of infected contacts per index case, not the offspring distribution. This has been clarified in the legend. Therefore, the overdispersion parameter described in the legend can be interpreted as directly informing

the quantity $P(\text{number transmissions} \geq 2 \mid \text{number transmissions} \geq 1) * P(\text{siblings are contacts of each other})$.

The wording has been clarified in the Discussion section: "This explanation is supported by the fact that an index case with an infected contact was more likely to have additional infected contacts (Supplementary figure 4) and by previous reports on the role of super-spreading in COVID-19 transmission."

"backward traced contacts seem to consist mostly of sibling rather than parent cases,
also supports this mechanism (Figure 5). Fourth, more distant relatives in the
transmission tree

could be detected due to wider circulation in an index case's social circle."

The inference that these contacts are mostly siblings was implicitly based on the assumption that they are either siblings, children, or parents: nothing else. If the authors consider that these contacts could be elsewhere in the transmission tree, for accuracy they should not be described as "most likely siblings" but "more likely to be siblings than parents or children".

The phrasing was changed as follows: "The finding that backward traced contacts seem to more likely consist of sibling rather than parent or child cases, also supports this mechanism".

"These results speak in favour of simply referring all close contacts in the extended
tracing

window for testing and quarantine, even if they were not present at the suspected
source event.

Additionally, we show that jurisdictions favouring the implementation of a source
investigation

strategy would do well to switch to an extended contact tracing window approach
when no

clear source event is identified at the time of the contact tracing interview"

I suggest replacing "Additionally, we show that" by "Alternatively", because the two recommendations differ: the first recommends not bothering with source investigation, and the second recommends how to do it if you do it.

This was changed as suggested.

447 "Third, the population was almost entirely unvaccinated during the main
448 study period."

It might be worth stating why this is a limitation - is extrapolating from these findings to a vaccinated population just adding some unknown unknowns, or is a bias expected in one direction?

We changed the phrasing as follows:

“Third, the population was almost entirely unvaccinated during the main study period.

Whether the influence of mass vaccination is different for backward versus forward contact tracing remains unclear and merits further study.”

597: comma for decimal separator

This was corrected.

"Contacts were assigned to either the standard tracing window group, a reference group
mirroring standard practice, or to the extended tracing window group, based on when their last
close contact with the index case took place."

I suggest enclosing "a reference group mirroring standard practice" with either dashes or parentheses, because commas make this seem like there are three groups instead of two.

The punctuation was changed as suggested.

"In the last period, with the Omicron strain dominant, contacts who did
not develop symptoms or undergo testing in the 7 days after exposure were considered "not
infected"."

Was this outcome considered as lost-to-follow-up for previous viral variants? Why was it re-defined to not infected for Omicron?

We clarified the reasoning behind this decision: “In the last period, with the Omicron strain dominant, government-mandated testing for close contacts was abandoned, leading to very low testing rates. Therefore, contacts who did not develop symptoms or undergo testing in the 7 days after exposure were considered “not infected” during this period.”

With best wishes to the authors,

REVIEWER COMMENTS

Reviewer #3 (Remarks to the Author):

Apologies to the authors for the delay: I went on holiday the day before they resubmitted.

THE NEW BRANCHING PROCESS

Adding a new component to the paper - mathematical modelling - is admirable, though I felt the study was acceptably complete without it. The new work requires additional scrutiny, and currently I struggle to understand what was done. Including the underlying code is helpful for reproducibility, but the text should provide sufficient detail to understand the method. Firstly, why was this piece of modelling done? The answer seems to be only mentioned in supplementary:

Many of the reported contacts in our dataset were outside the study population, which means
their infected contacts were not iteratively traced using the same backward tracing strategy.
To estimate how efficient backward contact tracing would be if all infected contacts were
iteratively traced, we used a simple branching process model.

I suggest moving this motivation to the start of the main-text section about the model: it's helpful to know why a piece of work was done before reading about what its results were. If I've understood, this motivation implies the model would not have been necessary if all contacts were part of the study population: in that case, the intervention actually used coincides with the intervention we want to evaluate - iterative tracing for everyone - and one could have simply calculated how much transmission potential was averted for each of the observed cases, then taken an average of those values. It follows that, when fitting this model to the observations from only partially iterative tracing, one needs to discriminate between contacts that were part of the study population and contacts that weren't. However, there is no mention of this in the method, and so I get lost in trying to match what was done and the aim.

I also struggle to piece together the individual components of the model, which have already been described, into a whole: I think some description of the overall structure is needed. It's unclear to me even whether this branching process model is stochastic, as such models usually are, or deterministic. Using a pseudocode representation of it the model (which could be helpful in explaining the structure), I think I understand this much:

For every contact i in our data (perhaps only in the study population?):

Assign i to a group according to their tracing sequence

Calculate i 's number of quarantine days $q_i = \dots$

Calculate i 's number of tests $s_i = \dots$

If i is infected:

calculate i 's fraction of averted transmissions $a_i = \dots$

For every group X :

Calculate $g_X =$ number infected in group X

Calculate $c_X =$ total number in group X / number of infected contacts in group X -hat

Following this, I don't understand what set of operations were applied to the group-level and individual-level vectors calculated so far (q, s, a, g, c) to arrive at the overall results.

Does the branching process assume that no individual can be a contact of more than one case?

Once every contact in the dataset has been assigned to a group, then all the sequences that could be defined by appending either an F or a B to every sequence observed are implicitly empty. e.g. if we observed contacts traced via the sequence (F, B, F) but no one traced via (F, B, F, F) or (F, B, F, B), then the latter two groups effectively exist but with size zero. Does this affect the calculation? Should it? The following point may be related, I'm not sure:

259 If the

260 number of study cases in $X_{\{n\text{-hat}\}}$ was smaller than or equal to 5, c_X and g_X were assumed to equal

261 0.

Why do this? Why the threshold value 5? Might the result be sensitive to this ad hoc procedure?

The model assumes that tracing someone has perfect effectiveness in preventing transmission from the moment of tracing onwards, i.e. transmission after tracing is impossible. This assumption should

be stated.

The symbol "*", when used in typeset equations rather than in computer code, usually denotes convolution or other more exotic operations. The multiplication symbol "x" should be used here.

Throughout the section, X is used to denote both a particular tracing sequence (such as one forwards tracing step then one backwards one) and the group of contacts arising from the final step of that tracing sequence. It would be good to use more precise language, as in places it leads to ambiguity. For example here:

$G(X)$ is the number of infected contacts identified through tracing sequence X
and here:

Similarly, the total number of contacts in X is given by:

I first interpreted these as referring to contacts involved anywhere in the sequence X, whereas the intended meaning is only contacts in the final step of X.

with c_X the number of traced contacts according to a_n per case of the previous
generation $X_{\hat{n}}$.

I suggest clarifying to "with c_X the number of traced contacts per case of the previous generation $X_{\hat{n}}$, i.e. in the step a_n "

The total number of detected cases and contacts, not including the primary index case, across
all tracing generations, is given by G_{tot} .

G_{tot} thus defined is the total number of cases and _infected_ contacts, or equivalently the total number of infections detected. It excludes uninfected contacts.

The source of the transmission timing distribution I provided - Ferretti et al - is still only in pre-print form and so is best not described as "published".

OTHER MINOR COMMENTS

The authors have clarified that Supplementary Figure 4 shows the distribution not of (a) the number of contacts infected by an index case, but of (b) the number of contacts per index case who were infected (by anyone). However, in the figure legend and in Discussion (where the figure is cited), this result is interpreted as though it is (a), because it is (a) and not (b) that informs us of the extent of superspreading. Overdispersion in (b), say through a small number of cases having a large number of infected contacts, could be due to those cases starting a long one-to-one chain of transmission through a group of individuals all known to initial index case, not a short one-to-many burst of transmission (the definition of superspreading). Further analysis on this point would probably be out of scope, but the assumptions and reasoning in the interpretation should be laid out. If (a) and (b) cannot be easily related, perhaps the interpretation of this result should be dropped.

The new Figure 8, panel C, considers "cost per averted infectivity". I suspect "cost per averted infection" is the intended meaning: infectivity is usually synonymous with infectiousness, which has dimensions of per unit time i.e. a rate of infecting others. I also think the averted infection considers only the first generation of infections caused by each index case, i.e. not the whole averted chain of transmissions downstream of the index case, the size of which would depend on the model's time frame (e.g. allowing the epidemic to spread until herd immunity is reached). Perhaps this could be clarified.

380 This study lays out a strategy for backward contact tracing which markedly improves the
381 effectiveness of contact tracing in the setting of COVID-19. It identified an additional 42% (or
382 55% when taking iterative tracing into account)

I suggest clarifying that 42% was an empirical measurement, whereas 55% is the prediction from a mathematical model.

With best wishes to authors,
Chris Wymant

REVIEWER COMMENTS

Reviewer #3 (Remarks to the Author):

Apologies to the authors for the delay: I went on holiday the day before they resubmitted.

THE NEW BRANCHING PROCESS

Adding a new component to the paper - mathematical modelling - is admirable, though I felt the study was acceptably complete without it. The new work requires additional scrutiny, and currently I struggle to understand what was done. Including the underlying code is helpful for reproducibility, but the text should provide sufficient detail to understand the method. Firstly, why was this piece of modelling done? The answer seems to be only mentioned in supplementary:

222 Many of the reported contacts in our dataset were outside the study population, which means

223 their infected contacts were not iteratively traced using the same backward tracing strategy.

224 To estimate how efficient backward contact tracing would be if all infected contacts were

225 iteratively traced, we used a simple branching process model.

I suggest moving this motivation to the start of the main-text section about the model: it's helpful to know why a piece of work was done before reading about what its results were. If I've understood, this motivation implies the model would not have been necessary if all contacts were part of the study population: in that case, the intervention actually used coincides with the intervention we want to evaluate - iterative tracing for everyone - and one could have simply calculated how much transmission potential was averted for each of the observed cases, then taken an average of those values. It follows that, when fitting this model to the observations from only partially iterative tracing, one needs to discriminate between contacts that were part of the study population and contacts that weren't. However, there is no mention of this in the method, and so I get lost in trying to match what was done and the aim.

Thank you for this comment. The model indeed aimed to make a first step towards extrapolating the specific contact tracing approach in the study to the general population, by generalizing it to all contacts of the traced index cases regardless of whether they were part of the study population and thus iteratively traced. We have now specified this at the top of the corresponding section:

“As mentioned, iterative contact tracing of infected contacts is thought to play a larger role in backward contact tracing. However, many of the reported contacts in our dataset were outside the study population, which means their contacts were not iteratively traced using the same backward tracing strategy if infected. To estimate how efficient backward contact tracing would be if all infected contacts were iteratively traced, we used a simple branching process model.”

I also struggle to piece together the individual components of the model, which have already been described, into a whole: I think some description of the overall structure is needed. It's unclear to me even whether this branching process model is stochastic, as such models usually are, or deterministic. Using a pseudocode representation of it the model (which could be helpful in explaining the structure), I think I understand this much:

For every contact i in our data (perhaps only in the study population?):

Assign i to a group according to their tracing sequence

Calculate i 's number of quarantine days $q_i = \dots$

Calculate i 's number of tests $s_i = \dots$

If i is infected:

calculate i 's fraction of averted transmissions $a_i = \dots$

For every group X :

Calculate $g_X =$ number infected in group X

Calculate $c_X =$ total number in group X / number of infected contacts in group X -hat

Following this, I don't understand what set of operations were applied to the group-level and individual-level vectors calculated so far (q, s, a, g, c) to arrive at the overall results.

As suggested, we have added a pseudocode representation of the model in Supplementary Information. We have clarified that the model is deterministic.

Does the branching process assume that no individual can be a contact of more than one case?

Indeed, the same exclusion criterion applies here as described in the Methods section: "Individuals who were already identified as contacts exposed to a previously diagnosed index case within 7 days before the contact tracing interview were excluded as contacts from the second identified index case."

This was clarified in the description of the model: "The same exclusion criteria applied as in the rest of the study. For example, a contact of multiple cases was only included as a contact of the first case to report them."

Once every contact in the dataset has been assigned to a group, then all the sequences that could be defined by appending either an F or a B to every sequence observed are implicitly empty. e.g. if we observed contacts traced via the sequence (F, B, F) but no one traced via (F, B, F, F) or (F, B, F, B), then the latter two groups effectively exist but with size zero. Does this affect the calculation? Should it? The following point may be related, I'm not sure:

If the
number of study cases in $X_{\{n\text{-hat}\}}$ was smaller than or equal to 5, c_X and g_X were assumed to equal
0.

Why do this? Why the threshold value 5? Might the result be sensitive to this ad hoc procedure?

Iterative contact tracing of infected contacts logically stops after a certain number of tracing generations, because the number of infected contacts that can be detected is limited by the number of cases in the cluster. This assumes that generally, the time from detection of an infected contact to detection of their iteratively traced infected contact is shorter than the generation interval of the transmission tree. In other words, contact tracing spreads through the social network faster than the disease.

This is also what we observed (Supplementary Figure 8): after a certain number of tracing generations, very few of the reported contacts were infected. Some tracing sequences which we did not observe, could be observed in other settings, possibly leading to different results. We see this as an expected phenomenon and a logical consequence of extrapolating empirical observations of tracing sequences.

As we determine the total number of expected infected contacts by multiplying the average number of infected contacts per case g_X for each tracing step in a sequence, the model can become sensitive to outliers when the number of observations to determine g_X is small. That is why we decided to discount groups with a very low number of observations. The effect of this decision is small, as by definition these groups only represent a small fraction of the total number of identified contacts.

We clarified this in the model description: "If the number of study cases in X_n was smaller than or equal to 5, c_X and g_X were assumed to equal 0. This was done to avoid the model becoming too sensitive to outliers when the number of observations to determine g_X was small."

The model assumes that tracing someone has perfect effectiveness in preventing transmission from the moment of tracing onwards, i.e. transmission after tracing is impossible. This assumption should be stated.

This was added to the model description in SI.

The symbol "*", when used in typeset equations rather than in computer code, usually denotes convolution or other more exotic operations. The multiplication symbol "x" should be used here.

This was adjusted.

Throughout the section, X is used to denote both a particular tracing sequence (such as one forwards tracing step then one backwards one) and the group of contacts arising from the final step of that tracing sequence. It would be good to use more precise language, as in places it leads to ambiguity. For example here:

245 $G(X)$ is the number of infected contacts identified through tracing sequence X
and here:

251 Similarly, the total number of contacts in X is given by:

I first interpreted these as referring to contacts involved anywhere in the sequence X , whereas the intended meaning is only contacts in the final step of X .

We clarified this ambiguity in the model description by denoting the sequence as X and denoting the contacts in the last step of this sequence as the group with sequence X .

253 with c_X the number of traced contacts according to a_n per case of the previous
254 generation $X_{\hat{n}}$.

I suggest clarifying to "with c_X the number of traced contacts per case of the previous generation $X_{\hat{n}}$, i.e. in the step a_n "

We changed this in the text.

255 The total number of detected cases and contacts, not including the primary index case, across
256 all tracing generations, is given by G_{tot} .

G_{tot} thus defined is the total number of cases and infected contacts, or equivalently the total number of infections detected. It excludes uninfected contacts.

This was changed as suggested.

The source of the transmission timing distribution I provided - Ferretti et al - is still only in pre-print form and so is best not described as "published".

We removed the term "published".

OTHER MINOR COMMENTS

The authors have clarified that Supplementary Figure 4 shows the distribution not of (a) the number

of contacts infected by an index case, but of (b) the number of contacts per index case who were infected (by anyone). However, in the figure legend and in Discussion (where the figure is cited), this result is interpreted as though it is (a), because it is (a) and not (b) that informs us of the extent of superspreading. Overdispersion in (b), say through a small number of cases having a large number of infected contacts, could be due to those cases starting a long one-to-one chain of transmission through a group of individuals all known to initial index case, not a short one-to-many burst of transmission (the definition of superspreading). Further analysis on this point would probably be out of scope, but the assumptions and reasoning in the interpretation should be laid out. If (a) and (b) cannot be easily related, perhaps the interpretation of this result should be dropped.

As suggested, we have dropped the interpretation of Figure 4 as indicative of superspreading.

In the figure legend, the following sentence was removed: "These results suggest a degree of superspreading similar to what has been described in literature."

In the Discussion section, part of the sentence was removed as follows: "This explanation is supported by the fact that an index case with an infected contact was more likely to have additional infected contacts (Supplementary figure 4) and by previous reports on the role of super-spreading in COVID-19 transmission^{20,22,23}."

The new Figure 8, panel C, considers "cost per averted infectivity". I suspect "cost per averted infection" is the intended meaning: infectivity is usually synonymous with infectiousness, which has dimensions of per unit time i.e. a rate of infecting others. I also think the averted infection considers only the first generation of infections caused by each index case, i.e. not the whole averted chain of transmissions downstream of the index case, the size of which would depend on the model's time frame (e.g. allowing the epidemic to spread until herd immunity is reached). Perhaps this could be clarified.

"Averted infectivity" was calculated as the average estimated fraction of next-generation transmissions averted by isolating/quarantining a case detected through a certain iterative tracing sequence, multiplied with the modelled number of cases detected according to that sequence. Any averted transmissions in subsequent branches of the transmission tree are not included. In figure 8, panel a and c, which compare the cost and cost/benefit ratio respectively of backward relative to forward tracing, "averted infectivity" can indeed be considered equivalent to "averted infections", with the important caveat that this only includes the first subsequent generation of infections. We avoided the term "averted infections" in this figure because it could incorrectly be interpreted as all infections averted by averting entire branches of the transmission tree.

This has been clarified in the legend as follows: "'Averted infectivity' denotes the number of detected cases, multiplied with their remaining fraction of transmission potential according to Figure 5, panel a. This measure of benefit accounts for the observation that backward traced cases were detected later in their infectious cycle. In this figure, "averted infectivity" can be considered equivalent to the number of averted infections, with the important caveat that it only includes child cases of a detected case, not any subsequent averted branches of the transmission tree."

This study lays out a strategy for backward contact tracing which markedly improves the
effectiveness of contact tracing in the setting of COVID-19. It identified an additional 42% (or
55% when taking iterative tracing into account)

I suggest clarifying that 42% was an empirical measurement, whereas 55% is the prediction from a mathematical model.

We have clarified this sentence as follows: "It identified an additional 42% (or 55% in a mathematical model of iterative tracing) [...]"

With best wishes to authors,
Chris Wymant